# Uncertainties in the land use flux resulting from land use change reconstructions and gross land transitions

Anita D. Bayer[1], Mats Lindeskog[2], Thomas A.M. Pugh[3,1], Peter Anthoni[1], Richard Fuchs[4,5], Almut Arneth[1]

[1]Karlsruhe Institute of Technology KIT, Institute of Meteorology and Climate Research, Atmospheric Environmental Research, 82467 Garmisch-Partenkirchen, Germany.

[2]Department of Physical Geography and Ecosystem Science, Lund University, 223 62 Lund, Sweden.

[3]School of Geography, Earth & Environmental Science and Birmingham Institute of Forest Research, University of Birmingham, B15 2TT, United Kingdom.

[4]Wageningen University, Laboratory of Geoinformation Science and Remote Sensing, 6708PB Wageningen, the Netherlands.

[5]Environmental Geography Group, Department of Earth Sciences, Vrije Universiteit Amsterdam, De Boelelaan 1087, 1081 HV Amsterdam, the Netherlands.

Correspondence to: Anita D. Bayer (anita.bayer@kit.edu)

**Abstract.** Land-use and land-cover (LUC) changes are a key uncertainty when attributing changes in measured atmospheric $CO_2$ concentration to its sinks and sources, and must also be much better understood to determine possibilities for land-based climate change mitigation, especially in the light of human demand on other land-based resources. On the spatial scale typically used in terrestrial ecosystem models (0.5 or 1 degrees) changes in LUC over time periods of a few years or more can include bi-directional changes on the sub-grid level, such as the parallel expansion and abandonment of agricultural land (e.g. in shifting cultivation), or cropland-grassland conversion (and vice versa). These complex changes between classes within a gridcell have often been neglected in previous studies, and only net changes of land between natural vegetation cover, cropland and pastures accounted for, mainly because of a lack of reliable high-resolution historical information on gross land transitions, in combination with technical limitations within the models themselves. In the present study we applied a state-of-the-art dynamic global vegetation model with a detailed representation of croplands and carbon-nitrogen dynamics to quantify the uncertainty in terrestrial ecosystem carbon stocks and fluxes arising from the choice between net and gross representations of LUC. We used three frequently applied global, one recent global and one recent European LUC datasets, two of which resolve gross land transitions, either in Europe or in certain tropical regions. When considering only net changes, land-use-transition uncertainties (expressed as one standard deviation around decadal means of four models) in global carbon emissions from LUC ($E_{LUC}$) are ±0.19, ±0.66 and ±0.47 Pg C $a^{-1}$ in the 1980s, 1990s and 2000s, respectively, or between 14 % and 39 % of mean $E_{LUC}$. Carbon stocks at the end of the 20th century vary by ±11 Pg C for vegetation and ±37 Pg C for soil C due to the choice of LUC reconstruction, i.e. around 3% of the respective C pools. Accounting for sub-grid (gross) land conversions significantly increased the effect of LUC on global and European carbon stocks and fluxes, most noticeably enhancing global cumulative $E_{LUC}$ by 33 Pg C (1750-2014) and entailing a significant reduction in carbon stored in vegetation, although the effect on soil C stocks was limited. Simulations demonstrated that assessments of historical carbon stocks and fluxes are highly uncertain due to the choice of LUC reconstruction and that the consideration of different contrasting LUC reconstructions is needed to account for this uncertainty. The analysis of gross, in addition to net, land-use changes showed that the full complexity of gross land-use changes is required in order to accurately predict the magnitude of LUC change emissions. This introduces technical challenges to process-based models and relies on extensive information on historical land use transitions.

Keywords: land-use change, gross land transitions, land-use flux

## 1. Introduction

Under the increasing demand of a growing population for food and fiber, as well as for bioenergy, greater anthropogenic pressures on the global land area are expected. Today, carbon dioxide ($CO_2$) emissions resulting from land-use and land-cover (LUC) change are the second largest contributor to anthropogenic emissions to the atmosphere after fossil fuel combustion (Le Quéré et al., 2015a), and they are a term that is associated with large uncertainties. LUC and their changes include the processes when land is converted from one land-cover type to another (e.g. the conversion of forest to cropland or grasslands to pastures), and the effects from LUC related to the management of the land, such as e.g. cropping practices, fertilizer use, irrigation and different types of tillage. LUC changes affect the cycling of carbon (C), energy, water and other nutrients (phosphorous, nitrogen), in many cases enhancing greenhouse gas (e.g. $CO_2$, $N_2O$, $CH_4$) emissions from agricultural soils and pastures when compared to natural vegetation and altering species composition. These changes go hand-in-hand with altered characteristics such as surface albedo, surface aerodynamic roughness and rooting depth (Pongratz et al., 2010).

Conversions from natural vegetation to croplands and pastures generally reduce C stored in vegetation (Baccini et al., 2012), decrease soil C stocks in croplands but not in pastures (Guo and Gifford, 2002; McLauchlan, 2006) and, unless fertilized, reduce soil nitrogen (N) pools (McLauchlan, 2006). The alteration of C and N pools is mainly a result of initial deforestation, and of the decreased litter input due to biomass extraction upon harvest and accelerated soil decomposition rates. The latter being stimulated through management practices such as tillage or a changed microclimate at the soil surface. However, in some regions croplands show increased C sequestration potential compared to the natural vegetation owing to enhanced growth under improved agricultural practices including fertilization and irrigation (Ciais et al., 2010; Schulze et al., 2010). Legacy fluxes can change C due to e.g., an imbalance between reduced litter input and decomposing dead biomass and affect LUC emissions over decades or more (Gasser and Ciais, 2013; Houghton, 2010; Krause et al., 2016; Pugh et al., 2015). During vegetation recovery on abandoned agricultural land, secondary land ecosystems sequester C due to regrowing vegetation and re-accumulation of C in soils. These LUC-related processes determining regional sources and sinks of C entailed a global total net C flux to the atmosphere over the past centuries (deB Richter and Houghton, 2011; Houghton et al., 2012; McGuire et al., 2001; Le Quéré et al., 2015a).

A number of studies have recently highlighted the importance of different definitions when assessing the net carbon flux from LUC ($E_{LUC}$) related to the fact that in different studies different LUC component-fluxes are included in the overall $E_{LUC}$ calculation (Gasser and Ciais, 2013; Pongratz et al., 2014). Likewise, it is important to consider whether or not historical effects of environmental change are included in assessments of cleared C stocks as part of $E_{LUC}$. Less focus so far has been put on the explicit datasets of historical land use employed (Le Quéré et al., 2013). A limited number of historic LUC reconstructions are available at global scale, mostly at 0.5° spatial resolution (Hurtt et al., 2016, 2011; Kaplan et al., 2012; Klein Goldewijk, 2016; Klein Goldewijk et al., 2011; Olofsson and Hickler, 2008; Pongratz et al., 2008; Ramankutty and Foley, 1999), two of which are very similar (datasets of Hurtt et al. and Klein Goldewijk et al. are consistent when the corresponding versions are compared). At continental scale some higher resolution reconstructions exist for instance for Europe (Fuchs et al., 2015b; Kaplan et al., 2009; Williams, 2000). Most reconstruction approaches combine information on current and recent historical LUC from national statistics with estimates of global population distribution and growth as the main driver of historical LUC. Model assumptions are made to fill data gaps and extrapolate the available information to create subnational patterns, and therefore large uncertainties arise both from the original data sources and modeling assumptions (see, e.g. Klein Goldewijk and Verburg, 2013). However, reconstructions on continental scales are able to use a more data-driven approach (e.g. Fuchs et al., 2013, 2015c) compared to global land reconstructions, since the data availability is often better for these study areas.

Most historical LUC reconstructions focus on the difference in net area under natural, cropland or pasture vegetation cover in a grid location between two time steps (net land changes) instead of explicitly showing the sum of the absolute value of all land transitions occurring on a sub-grid scale (gross land changes). In particular over coarser grid-resolutions, gross land-cover changes allow a deeper view of LUC, tracking land conversion events such as the parallel expansion and abandonment of agricultural land, e.g. as in shifting cultivation (cycle of cutting forest for agriculture and abandoning it after some years of usage, followed by a period of fallow with

regrowing forests). This entails altered biogeochemical dynamics within different sub-sections of a gridcell, e.g. secondary land acts as a C sink during vegetation regrowth, while additional land clearing leads to relatively rapid loss of C stocks in vegetation and soils, along with other changes in vegetation composition, nutrients and biogeophysical properties (e.g. Houghton et al., 2012). Accounting in ecosystem models separately for the effects of individual transitions, e.g., 10% of an area converted from natural vegetation to cropland while another 10% of cropland is abandoned for regrowth over the same time period, will therefore lead to a very different response of ecosystem states and fluxes compared to the effects of net changes, which in this case would be zero.

The availability of land-use information including gross land transitions is limited due to a lack of reliable historical information to determine them. However, a few data sets exist representing gross land transitions, such as the global dataset by Hurtt et al. (2011) (recently updated, see Hurtt et al., 2016) who provide model estimates of shifting cultivation in certain tropical areas based on a map of Butler (1980) and assumed land rotation rates. Fuchs et al. (2015b) recently estimated gross land changes for Europe over the 20[th] century based on empirical evidence. As the number of gross land transitions can greatly exceed the number of net transitions at spatial resolutions typically employed for global studies, neglecting these can lead to a serious underestimation of LUC dynamics with implications for biogeochemical, ecological and environmental assessments (Fuchs et al., 2015a, 2015b; Stocker et al., 2014; Wilkenskjeld et al., 2014). Earlier studies revealed significant differences when ecosystem C dynamics were simulated when accounting for gross land changes in areas of shifting cultivation in addition to net changes as specified by Hurtt et al. (2011) (e.g. Shevliakova et al., 2013; Stocker et al., 2014; Wilkenskjeld et al., 2014). Others have implemented their own assumptions on spatial distribution and rotation scheme under shifting cultivation and combined these with C-cycle calculations (Olofsson and Hickler, 2008; Stocker et al., 2014).

In this study we use a state-of-the-art dynamic global vegetation model (DGVM) to calculate ecosystem C stocks and fluxes in response to different LUC reconstructions, (1) to explore the consistency of different LUC representations and to quantify the uncertainty in ecosystem C stocks and fluxes, including $E_{LUC}$, resulting from the different reconstructions, and (2) to quantify the effect of accounting for gross land transitions in addition to net changes in LUC. We use five historical LUC reconstructions, four of which are global and one only for the European domain. One of the global datasets represents gross transitions due to shifting cultivation in certain tropical regions, and the European dataset represents gross transitions from all sources in Europe. We apply a model with representation of LUC and changes therein, including a number of crop functional types and C-N dynamics in natural vegetation and crops. We exclude wood harvest as a form of forest management that can be represented as gross land transitions from our analysis as, although national data on wood harvest are available, its parameterization in models is poorly constrained on a global scale, e.g. the effects strongly depend on assumptions on the harvest type (clear cut, selective logging, or a mixture of both), or assumptions regarding turnover times of harvested C (Wilkenskjeld et al., 2014).

## 2. Methods
### 2.1. Land-use datasets

For the global scale, three historical LUC datasets were selected that are frequently used for ecosystem modeling studies, as well as one recently released dataset that is expected to be frequently used in the future. These datasets run at least from 1700 to present and are also the basis for future LUC scenarios (e.g. van Asselen and Verburg, 2013; Hurtt et al., 2011). For Europe we additionally considered one recently published dataset running from 1900-2010. Table **1** provides an overview of the LUC datasets and their characteristics.

Ramankutty and Foley (1999) (RAMA) published changes in cropland area for the period 1700 to 1992 in 5 min global resolution. The dataset was built based on historical cropland inventory data at national and subnational levels in combination with a remote-sensing derived cropland map for 1992. The algorithm to hindcast LUC distributed the historical cropland area within political units, i.e. the 1992 cropland areas are scaled for each political unit so that the cropland total matches historical inventory data. Therefore, the reconstructed changes in historical croplands are consistent with the history of human settlement and patterns of economic developments, although they do not resolve changes in LUC dynamics below the smallest spatial entity in the inventory data.

The analysis was revised in 2012 so that it also accounts for pasture areas and the dataset was extended until 2007 in 0.5° x 0.5° spatial resolution (Ramankutty, 2012). Natural areas are calculated as the remainder of cropland and pasture areas. Here we apply the revised 0.5° x 0.5° version.

The History Database of the Global Environment (HYDE) 3.1.1 (Klein Goldewijk et al., 2010, 2011) provides spatially gridded maps of cropland and pastures at 5 min resolution for the period 10 000 BC to AD 2000. Here,

historical population data and national and sub-national statistics of change in agricultural area (mainly the United Nations Food and Agriculture Organization, FAO, supplemented with numerous other statistics) were combined to a land-use per capita relationship. Land use was then allocated for present day based on satellite-derived land cover for 2000 and for the past by a combination of this base map with a number of weighting and suitability factors such as population density and habitat information on soil suitability, distance to rivers, slopes

etc. Temporal resolution is 10 years for the historical period after 1700 and annual after 2000. The HYDE dataset used here was extended until 2005 (Klein Goldewijk et al., 2015), and later until 2013 in the 2014 global carbon budget analysis (Le Quéré et al., 2015a).

The Hurtt et al. (2011) Land Use Harmonization v1 (LUH1) database is based on the land-use data of HYDE (Klein Goldewijk et al., 2010, 2011) for the historical period (1500-2005), and combines these with national

statistics of historical wood harvest and assumptions regarding shifting cultivation in certain tropical regions. Additional assumptions were made regarding the prioritization of land for conversion and logging, the wood harvest spatial patterns and the residence time of land in agricultural use in shifting cultivation areas. LUH1 data provide fractional data on cropland, pasture, primary and secondary vegetation as well as gross transitions between land-use states based on shifting cultivation on 0.5° x 0.5° spatial resolution. Secondary land is defined

as natural land that was previously used for agriculture and is recovering from this disturbance. Shifting cultivation is implemented as bi-directional LUC change with an assumed rotation period of 15 years, corresponding to an annual turnover rate of 6.7 % of the area (Hurtt et al., 2011). Although the history of shifting cultivation is not known, it is today present mainly in tropical regions and therefore in the LUH1 dataset it is limited to certain tropical regions for the historical period (Fig. S1**Error! Reference source not found.**,

Olofsson and Hickler, 2008). The LUH1 dataset was extended until 2014 for the 2015 global carbon budget analysis (Le Quéré et al., 2015b), using an early version of HYDE 3.2 as the basis (now published in final version as Klein Goldewijk, 2016) and following the same methodology as described in Hurtt et al. (2011). The version of LUH1 used in this study is therefore a more recent development than that used for CMIP5 experiments (Taylor et al., 2012), but an earlier version than the very recent LUH2 release (Hurtt et al., 2016).

As LUH1 is a modeled product that is based on the underlying HYDE database, these products are very similar when the corresponding versions of each dataset are considered (Hurtt et al., 2011). Note that the version of HYDE used for our study (version 3.1.1, see above) is not the same as the version of HYDE that underlies the LUH1 data used here (early version of HYDE 3.2). HYDE 3.1.1 and 3.2 differ in terms of driving population data and the algorithms used (Klein Goldewijk et al., 2016). The HYDE and LUH1 data used in this study

therefore differ in both their spatial and temporal distribution of land-use fractions (Fig. 1; Klein Goldewijk et al., 2016).

In addition, we used the very recent release of the Land Use Harmonization v2 (LUH2, Hurtt et al., 2016) that was developed for CMIP6 intercomparison project (Eyring et al., 2016). This global dataset covers the period 850-2015 and follows the same methodology as LUH1 described above, but uses HYDE version 3.2 (Klein

Goldewijk, 2016) as a basis, along with updated attributes on wood harvest and shifting cultivation, a higher spatial resolution of 0.25°, more detailed land-use transitions and additional land management information (such as irrigation and fertilizer use). From the LUH2 dataset, only LUC states and net land transitions aggregated to a spatial resolution of 0.5° x 0.5° were considered in this study, as a very new dataset that very likely will frequently be applied for modeling studies in the future. As LUC patterns of LUH2 are directly based on

HYDE3.2, these two datasets are very similar in their land use information (data not shown) and therefore resulting ecosystem C stocks and fluxes are expected to be very similar. The detailed LUC gross transitions provided by LUH2 could not be preprocessed in time to be used here.

The only non-global LUC dataset considered here, the HIstoric Land Dynamics Assessment (HILDA; Fuchs et al., 2013, 2015b), reconstructs gross and net land changes for the EU27 (EU from 2007-2013) plus Switzerland

at 1 km spatial resolution (Fig. S1). Net land conversions are based on national statistics. To account for gross changes, empirical evidence (mostly time series of large-area LUC maps and national surveys) on historic gross LUC changes was aggregated to derive an overall gross/net ratio per LUC class and a relative weighted land conversion matrix which were applied to national net change data. The allocation of LUC fractions to grid cells was done based on probability maps for each LUC class, forest volume stock maps and large-scale historic LUC maps (Fuchs et al., 2015c). An aggregated version of the CORINE 2000 land-cover dataset was used as base map for the year 2000. The initial LUC dataset that was built based on empirical evidence covers 1950-2010 in decadal steps but was extrapolated back to 1900 to assess the long-term impacts of changes in LUC. For each time step the 1 km grid cells were classified as being dominated by settlement, cropland, forest, grassland (incl. managed pastures), other land (glaciers, sparsely vegetated areas, beaches and water bodies) or water. Here, we consider only the gross LUC reconstruction of HILDA (Fuchs et al., 2015b) and derive net LUC changes from gross land transitions. In the original HILDA net LUC reconstruction (Fuchs et al., 2015b) the results differ spatially from our net reconstruction due to the use of different land allocation mechanisms under net and gross changes in their analysis.

## 2.2. LPJ-GUESS ecosystem model

We use the LPJ-GUESS DGVM (Sitch et al., 2003; Smith et al., 2001) with updates for land-use change (Lindeskog et al., 2013) and C-N coupling in natural vegetation and crops (Olin et al., 2015; Smith et al., 2014) allowing for the simulation of nitrogen limitation on plant and crop development. Three distinct land-use types are used (natural vegetation, pasture and cropland) with natural vegetation modeled by 9 woody and 2 grass plant functional types (PFTs) (as in Smith et al., 2014), which are distinguished in terms of their bioclimatic preferences, photosynthetic pathways and growth strategies. Vegetation structure, dynamics and competition between age cohorts of a PFT population are explicitly represented in the LPJ-GUESS model. Croplands are represented by three crop functional types (CFTs) that are parameterized using information on summer wheat, winter wheat and maize, and with crop-specific processes including dedicated carbon allocation and phenology, explicit sowing and harvest representation, irrigation, fertilization and cover crops as grass growing in between cropping periods (Olin et al., 2015). Pastures are modeled using competing C3 and C4 grass PFTs, where each year 50 % of the C and 12.5 % of the N in above-ground biomass was removed as a representation of grazing (Krause et al., 2016; Lindeskog et al., 2013).

In LPJ-GUESS, upon conversion of natural land to cropland and pastures, the natural vegetation is cleared and 97% of wood (of which stem wood 65%, branches and coarse roots 32%) and 71% of leaf biomass is harvested. Out of the harvested stem wood, one third goes to a product pool with a turnover time of 25 years. The rest of the harvested biomass is oxidized and released to the atmosphere, while the remaining biomass enters the litter pool (see Lindeskog et al., 2013). In reductions of the natural vegetation area, young stands (but older than 15 years, the assumed rotation period in shifting cultivation by Hurtt et al., 2011) are converted before older stands. Following agricultural abandonment, natural vegetation recolonizes the land in a typical succession from herbaceous to woody plants, if environmental conditions are suitable for tree growth. Competition for resources and light among age cohorts of woody PFTs is simulated directly through gap dynamics (see, e.g. Bugmann, 2001).

The model has been evaluated extensively and has demonstrated skills in capturing large-scale vegetation patterns (Hickler et al., 2006, 2012) and dynamics of the terrestrial carbon cycle (Ahlström et al., 2012; Morales et al., 2005; Olin et al., 2015; Piao et al., 2013; Pugh et al., 2015; Smith et al., 2014). The carbon flux response was close to the ensemble mean in a recent intercomparison of nine dynamic global vegetation models (Sitch et al., 2015).

## 2.3. Simulation protocol

LPJ-GUESS was run at 0.5° x 0.5° resolution with simulations beginning in year 1700. CRU TS 3.21 historical global climate data (University of East Anglia Climatic Research Unit (CRU), 2013) was used for the period 1901-2014. 1700-1900 climate data was provided by repeating 1901-1930 CRU climate with de-trended temperature data. Climate data for 2014 were repeated for the year 2015. Atmospheric $CO_2$ forcing was provided from observations from ice-cores and, later in the 20[th] century, atmospheric measurements (Tans and Keeling, 2015), with a value of 286.4 ppmv used from 1700 until the beginning of this dataset in 1860, and a final

atmospheric mixing ratio of 399.0 ppmv in 2015. Simulations were spun-up for 500 years using land-use fractions and $CO_2$ mixing ratio from the first simulation year, and de-trended climate data of the first 30 simulation years, with a longer spin-up for soil carbon pools (see Smith et al., 2014). Model spin-up was therefore identical for net and gross land changes. In order to assign cropland areas to crop functional types, global crop cover fraction was partitioned based on Portmann et al. (2010), and mapped to LPJ-GUESS crop

types, as described in Olin et al. (2015). Crop type fractions were held constant throughout the simulations. Where cropland was expanded into a hitherto un-cropped grid cell, average CFT fractions of the nearest neighboring cropland cells were used to populate it. Past values of global nitrogen deposition was taken from simulations from Lamarque et al. (2010 and 2011) and nitrogen fertilization of crops was estimated as in Zaehle et al. (2011). LPJ-GUESS simulations are summarized in Table 2.

Global simulations starting in 1700 were carried out using the four net and one gross LUC dataset (LUH1 net, RAMA net, HYDE net, LUH2 net, LUH1 gross), and for Europe starting in 1900 using two net and one gross LUC dataset (HILDA net, LUH1 net, HILDA gross). For these, all LUC input data were aggregated to 0.5° spatial resolution and decadal HILDA, HYDE and LUH1 LUC data were interpolated linearly to annual time steps. LPJ-GUESS uses annual land use states of the classes cropland, pasture, natural vegetation and barren land

(no vegetation, e.g. water or ice covered) as input for net LUC runs, that are complemented for gross LUC runs by annual gross transitions for each combination of two land-use classes. Land-use states of RAMA, HYDE and LUH2 were used directly. To generate net transitions from LUH1, annual land-use states were derived from land use states in 1700 and gross transitions from 1700 to 2014. HILDA land-use matrices providing land-use states and transitions together in form of an integer land-use category were translated to annual land-use states and

gross transitions for each combination of two land-use classes. Although some of these LUC products represent changes between forested and non-forested land, in the simulations done here, only the changes between the classes croplands, pastures (FAO category 'permanent pasture'), natural vegetation and barren land (available for LUH1, LUH2 and HILDA) were considered; the composition of natural vegetation was simulated directly by LPJ-GUESS. Primary and secondary vegetation as in LUH1 and LUH2 (wood harvest), and the forest class in

HILDA and LUH2 were represented by natural vegetation. The HILDA LUC classes of settlements, water and other land were aggregated to the LUC class barren. The grassland class in HILDA comprises both pastures, meadows and natural grasslands, but was used for pastures, a reasonable assumption for Europe due to the small area of truly natural, unmanaged grassland in Europe (Wilkins et al., 2003). For global simulations a land mask was used that includes only cells of the ice-free land area for which all four global LUC datasets provide data

(58 790 cells). For the Europe, all 0.5° grids that contained at least one HILDA cell were used (2 486 cells).

     We examine differences caused by the different LUC reconstructions on net land-use flux ($E_{LUC}$), deforestation flux, net primary productivity (NPP), and ecosystem C stocks in vegetation and soils. $E_{LUC}$ is calculated as the difference between a model simulation with transient historical LUC change (gross or net LUC change) and a simulation with constant LUC distribution as in the first simulation year (Table 2, LUC fixed to 1700/1900). All

simulations are driven by varying climate, atmospheric $CO_2$ mixing ratio, N deposition and N fertilization (see, e.g. Le Quéré et al., 2015; Pongratz et al., 2014). This method includes effects of LUC and changes therein interacting with climate and atmospheric $CO_2$, and is consistent with definition 1 of Gasser and Ciais (2013) and D3 of Pongratz et al. (2014). In the same way, the effect of accounting for LUC on NPP and C stocks in vegetation and soils was calculated as the difference between the simulation with gross or net LUC changes and

the respective reference simulation. Soil C includes both C in soils and litter. The deforestation flux is the C released upon land conversion only. In the calculation of net cumulative $E_{LUC}$ for global simulations the first 50 simulation years were ignored because of high carbon fluxes in the first decades of gross simulations, which reflected a re-equilibration under LUC including gross transition rates (i.e. shifting cultivation) that were not part of model spin-up, and effectively reflect emissions from shifting cultivation that occurred before 1700 (see, e.g.,

Stocker et al., 2014). Because gross transitions in Europe do not follow a systematic rotation such as shifting cultivation areas in the global simulations, this effect is not so directly applicable or apparent here and cumulative $E_{LUC}$ was calculated starting in 1900. We restrict our analysis of the effects of using different LUC reconstructions to the global scale, which has direct relevance for the global carbon budget; a detailed analysis of regional differences and processes is beyond the scope of this study.

## 3. Results

### 3.1. Historical land transitions

The most pronounced change in global vegetation cover over the historical period is the deforestation of natural areas for conversion into croplands and pastures (Fig. 1a), progressing at fairly low rates during the first decades after 1700, followed by a steadily increasing trend from about 1860 until a slow-down and stabilization sets in after about 1960. Total land area transitions before 1850 (Fig. 1c) are below 100 000 km$^2$ a$^{-1}$, from which they steadily increase with a rate of additional ca. 6 100 ± 1 100 km$^2$ a$^{-1}$ under transition each year (average of four LUC datasets 1850-1960, Table **3**). Transitions in all four LUC reconstructions peak between 1950 and 1960 when deforestation due to expansion of agriculture in the tropics and pasture expansion in grass and shrub dominated biomes was highest in the LUC reconstructions (Fig. S2). After the 1960s all four LUC reconstructions assume continued deforestation in the tropics at a lower rate and reforestation in Europe and some parts in Northern America following the abandonment of agricultural areas. Transitions around 1960 are believed to be influenced by the LUC reconstruction process, when model assumptions for the historical period before 1960 are merged with the records of the Food and Agriculture (FAO) records available thereafter.

The four global net LUC datasets applied here differ primarily in the total area of pasture and natural vegetation and in the regions in which these are located; RAMA and LUH2, in terms of global absolute area under natural LUC, lie in between LUH1 (lowest area under natural vegetation before 1960s) and HYDE (highest area under natural vegetation before 1960s, see Fig. 1a). Spatial patterns are generally more similar between LUH1, HYDE and LUH2 than with RAMA. However, the lower amount of natural areas and higher amount of pastures of LUH1 in the southern parts of Russia compared to the other three reconstructions is noteworthy. Other major differences between all four reconstructions occur in eastern Africa, eastern Europe and parts of the US and Canada (maps not shown). After 1960, LUH1, HYDE and LUH2 are very similar (showing major differences only in Australia). While the deforestation trend is shown by all four global LUC reconstructions, the absolute loss rates of natural vegetation differ, with HYDE being 14 % above the average of the other three LUC models (Table **3**). For present-day, differences are largest in natural areas and pastures, with RAMA reporting about 6 % more natural areas and about 18 % less pasture areas in 2007, compared to the other three reconstructions (Fig. 1a, Table **3**). Differences in pasture areas occur worldwide, but are somewhat higher in Saudi-Arabia, western China, Mongolia and Australia (maps not shown). Before 1950, differences in natural and pasture area between LUH1/LUH2/HYDE and RAMA exist predominantly in eastern Europe, southern parts of Russia and eastern Africa.

In Europe the two historical LUC reconstructions show the expansion of areas with regrowing natural vegetation after 1900 following land abandonment as a result of intensification on high production cropland (Fig. 1b). Net gain in natural regrowth area from 1900 to 2010 is about 60 000 km$^2$ (average of two LUC datasets, Table **4**). Rates of total land conversion in Europe over the first half of the 20$^{th}$ century (Fig. 1d) remain at a fairly constant level, with between 10 000 and 15 000 km$^2$ being converted each year. Rates of land conversion are only higher between 1950 and 1970 and after 1990.

The European land-use reconstruction HILDA shows the same trend in LUC over the historical period when compared with the same domain extracted from the global LUH1 product (Fig. 1b) but the two datasets disagree notably with respect to the absolute area of natural vegetation and pastures (Table **4**). HILDA shows substantially larger fractions of pasture than LUH1 especially in Scandinavia and southern Europe, while LUH1 shows higher pasture fractions than HILDA in central Europe and the Baltic area. The higher areas of pasture in HILDA may result from the fact that natural grasslands are included in the pasture class in HILDA but not in the pasture class of LUH1 (see Methods). The peak in total land conversion rates around mid of the century is shown in HILDA in two steps with slightly higher rates in 1950s and a maximum in the 1960s (13 700 and 19 100 km$^2$ per year) and in LUH1 in one step with a more than doubled rate in the 1950s compared to the previous decades (on average about 42 600 km$^2$ per year). From 1950 to 1960 the LUH1 dataset shows a rapid decrease in pasture area of 1.5 x 10$^5$ km$^2$ that is mainly reflected in a significant gain in natural areas.

### 3.2. Effects of different net LUC changes on carbon pools and fluxes

In LPJ-GUESS simulations all four global net LUC reconstructions resulted in similar projections of the land-use change flux $E_{LUC}$ as a source of C on the global scale, with the rate of emission accelerating from the early 1800s (Fig. 2a). Reflecting the time-series of the land transitions (Fig. 1c), $E_{LUC}$ peaked in the 1950s with emissions of about 2.0 to 2.6 Pg C a$^{-1}$. Mean $E_{LUC}$ was 1.2, 2.0, 2.1, and 1.3 Pg C a$^{-1}$ for LUH1, RAMA, HYDE and LUH2, respectively, at the end of the historical period (1998-2007, Table **3**) and cumulative $E_{LUC}$ since 1750 was between 199 Pg C for LUH2 and 229 Pg C for HYDE, respectively, in 2007 (Fig. 2b, Table **3**). From the four reconstructions, projections based on HYDE (with LUH2 being very close) resulted in the lowest emissions until the early 1900s, probably because of the lowest conversion of natural areas to pastures until this period compared to the other reconstructions (Fig. 1a). Also, when using the HYDE product, a second peak of $E_{LUC}$ of around 2.7 Pg C a$^{-1}$ occurred in the late 1990s (15-year average) that is not seen in LUH1, RAMA and LUH2 reconstructions (between 1.2 and 1.6 Pg C a$^{-1}$ in this period).

Global average NPP was simulated to increase strongly over the last century due to the effect of higher vegetation productivity under increased atmospheric $CO_2$ mixing ratio and (in cool areas) climate warming. Compared to reference simulations with LUC fixed in 1700 (red lines in Fig. 2d), all four LUC representations showed a reduced increase in NPP over the duration of the simulations (Table **3**, Fig. S3), with the reduction due to LUC at the end of the historical period (averages 1998-2007) being 1.9 Pg C for LUH1, 2.4 Pg C for HYDE, 3.3 Pg C for RAMA and 2.0 Pg C for LUH2 LUC reconstructions (Table **3**). Global total and time-series of NPP simulated with the four LUC reconstructions was very similar for RAMA and HYDE, and was about 2 Pg C lower for LUH1 and LUH2 simulations for the entire simulation period (Fig. 2d), possibly as a result of a higher pasture area instead of natural (woody) vegetation in the LUH1 and LUH2 data in the extra-tropical regions of Africa and areas in eastern Europe.

For global C stocks, both in vegetation and soils, the positive trend induced by $CO_2$ fertilized vegetation growth (red lines in Fig. 2e and f) was counteracted by the effects of LUC change. A minimum of vegetation C stocks during the simulation period was simulated for all LUC reconstructions in the 1960s when LUC reduced vegetation C stocks on average by 108 Pg C (Table **3**) compared to the reference simulations. Following the decline in conversion rates of natural into managed land thereafter, vegetation C stocks increased in response to vegetation productivity responding to the fertilizing effect of increasing atmospheric $CO_2$ concentration. Vegetation C stocks at the beginning of the simulation period were again similar for RAMA/HYDE and for LUH1/LUH2 reconstructions with about 482 Pg C, but on average about 31 Pg C lower for LUH1/LUH2 simulations because of the lower natural area in these datasets around 1700 (Table **3**).

Time-trends in soil C stocks over the simulation period followed similar trends as vegetation C stocks, albeit with a 5- to 10-year time lag (Fig. 2e, f). Loss in soil C as a result of accounting for LUC was a direct effect of the removal of biomass upon harvest reducing litter input in the following years and the increase in soil C decomposition rates associated with tillage. On average 73 Pg soil C were lost due to changes in LUC in the 2000s, with only a small variation of ±1 Pg C between LUH1, RAMA and HYDE reconstructions, and about 7 Pg C less for LUH2 (Table **3**, Table S3). Overall soil C stocks were again more similar for RAMA and HYDE (average 1 514 Pg C), and about 64 Pg C lower for LUH1 and LUH2 at the beginning of the simulation period (1700-1709, Fig. 2e, Table **3**).

In Europe, LUC caused emission of C until the 1960s, but turned into a sink thereafter (Fig. 3a, negative $E_{LUC}$) as a result of the reduction of pastures and also croplands in the second half of the last century and the regrowth of natural (woody) vegetation (Fig. 1b). This development is shown in simulations with both HILDA and LUH1, however the magnitude of the effect of LUC on C stocks and fluxes was somewhat stronger in simulations applying the LUH1 dataset, due to a higher deforestation rate in LUH1 until the 1950s (Fig. 3c, absolute land transitions were similar for both LUC datasets, Fig. 1d). To this sense, $E_{LUC}$ decreases from about 19 Tg C a$^{-1}$ (HILDA) and 38 Tg C a$^{-1}$ (LUH1) in the 1900s to about -52 Tg C a$^{-1}$ for HILDA and -80 Tg C a$^{-1}$ for LUH1 in the 2000s and cumulative $E_{LUC}$ from 1900 to 2010 was -936 Tg C for HILDA and -1 890 Tg C for LUH1 (Table **4**).

As in the global simulations, a positive trend in NPP from 1900 was also simulated for Europe, which is linked to increasing $CO_2$ fertilization (Fig. 3d). Accounting for changes in LUC reduced NPP in simulations applying

HILDA by 57 Tg C a$^{-1}$, but only slightly changed NPP in the 2000s when applying LUH1 (increase of 10 Tg C a$^{-1}$, Table **4**), because of highly productive croplands in central Europe (Fig. S4a). NPP simulated with HILDA was between 50 and 100 Tg C a$^{-1}$ lower over the simulation period than simulated with LUH1 (Table **4**). This also resulted from the lower share of natural areas in HILDA, as opposed to pastures, and thus lower productivity.

Similar trends in vegetation and soil C were simulated with both datasets, with changes dominated by deforestation over the first decades and reforestation thereafter (see Fig. 1b). In the 2000s vegetation C stocks were even higher under net LUC changes compared to the respective reference simulations (Table **4**). C stocks in vegetation of simulations using LUH1 were on average about 2 000 Tg C and 25 % higher than simulations applying HILDA (Fig. 3e, Table **4**). Differences in soil C stocks between the two LUC representations were small, with soil C being about 1 900 Tg C higher in LUH1 simulations (3 % of soil C stocks projected with HILDA) at the beginning of the simulation period (Table **4**). In comparison to the trend in C stored in vegetation, stocks in soils only increased from the 1950s on. Effects of LUC on C stocks in vegetation and soils were stronger for simulations applying LUH1 (Fig. 3e), showing increases in both C stocks in central and Eastern Europe but decreases in southern countries (Fig. S4b, c).

### 3.3.  Global and European effects of accounting for gross land transitions

The global land area undergoing LUC when considering gross land transitions based on the LUH1 dataset (Fig. 1c) was 4.7 times the net area converted (total transitions 1700-2014, Table **3**), with all additional land transitions in this dataset being generated by shifting cultivating in certain tropical regions (Fig. S1a). This increased the global land-use flux E$_{LUC}$ by about 0.14 Pg C a$^{-1}$ to 1.64 Pg C a$^{-1}$ at the end of the historical period (2005-2014 average flux) and resulted in cumulative E$_{LUC}$ being 33 Pg C higher in 2014 for gross compared to net LUC simulations (Fig. 2a, b, Table **3**). Global NPP was 1.5 Pg C a$^{-1}$ (i.e. 3 %) lower in simulations of gross land changes compared to the net simulations (Fig. 2d, Table **3**), which was an effect of lower mean levels of forest canopy closure in the tropical areas subject to shifting cultivation. For the same reason, the reduction of vegetation C stocks as an effect of accounting for gross effects was high, with 35 Pg C, i.e. -8 %, at the end of the simulation period. For soils, however, the effect was relatively low compared to the absolute size of soil C stocks, with an 11 Pg C reduction (-0.8 %, 2005-2014, Fig. 2e and f, Table **3**). The reduction of vegetation C stocks by the effects of LUC changes further increased by 24 % when accounting for gross LUC and for soil C stocks by 14 % (2005-2014, Table **3**).

For Europe, the HILDA gross dataset predicted land transitions (Fig. 1d) that were about 1.4 times higher when accounting for gross transitions relative to net LUC changes (total transitions 1900-2010, Table **4**) (see also Fuchs et al., 2015b) with significant gross land changes occurring all over Europe (Fig. S1b). As a result, gross E$_{LUC}$ was enhanced, compared to net, by about 11 Tg C a$^{-1}$ in the beginning of the simulation period (1901-1910). Cumulative gross E$_{LUC}$ was -531 Tg C in 2010, or 406 Tg C higher than E$_{LUC}$ under net LUC transitions (-936 Tg C), representing a reduced cumulative sink as the result of higher previous emissions from LUC when considering gross transitions (Fig. 3b, Table **4**). NPP was also lower in gross simulations, however differences were small (-18 Tg C a$^{-1}$, see Table **4**) and the gross simulation followed the same trend as the net LUC simulation. For C stocks on the European level, the difference between net and gross simulations was -158 Tg C for vegetation carbon and -254 Tg C for soil C stocks at the end of the simulation period (2001-2010, Fig. 3e and f, Table **4**).

### 4.  Discussion
### 4.1.  Uncertainties in carbon stocks and fluxes due to the choice of historical LUC reconstruction

Resulting from the fact that historical reconstructions of land use and its changes are inherently uncertain because of the limited existing data base that needs complementary assumptions (e.g. on land rotation times), it is widely acknowledged that a key uncertainty in estimating changes in C stocks and fluxes as a response to

LUC change stems from the choice of the LUC dataset (e.g. Houghton et al., 2012; Jain et al., 2013). With a detailed representation of succession when natural vegetation recolonizes abandoned agricultural lands, the representation of croplands by a number of crop functional types and the consideration of C-N interaction in natural vegetation and crops, the LPJ-GUESS model considers key processes and interactions that are crucial when accurate estimates of C stocks and fluxes are derived based on detailed dynamics on LUC (see, e.g.,

Hickler et al., 2004; Lindeskog et al., 2013; Olin et al., 2015; Pugh et al., 2015; Zaehle, 2013).

Uncertainties in $E_{LUC}$ resulting from the choice of LUC reconstruction (expressed as one standard deviation around decadal means), as quantified with the LPJ-GUESS model and the four global net LUC data sets, were ±0.19, ±0.66 and ±0.47 Pg C a$^{-1}$ (14 %, 39 % and 29 % of $E_{LUC}$) in the 1980s, 1990s and 2000s respectively (see Table **3** for exact periods). Among the four datasets, before the 1960s $E_{LUC}$ using LUH1 and RAMA were

similar, with the HYDE and LUH2 datasets differing somewhat from these as a result of regional differences in croplands and pastures, although global totals remained similar (e.g. HYDE showing less croplands in Northeastern US around 1900 than LUH1/RAMA, LUH2 showing similar cropland distribution as LUH1 but big differences in pastures in the southern parts of Russia, and LUH2 showing less pastures in southern Argentina and Kazakhstan than RAMA). These uncertainty calculations are consistent with estimates from previous studies

in which a subset of the three LUC hindcasts (LUH1, RAMA and HYDE, partially as earlier versions) were applied, sometimes in combination with a book-keeping method (Table S1). Uncertainties were also estimated for the 1980s as ±0.30 Pg C a$^{-1}$ from a synthesis experiment of Houghton et al. (2012), and for the 1990s as ±0.20 Pg C a$^{-1}$ found by Shevliakova et al. (2009) when using HYDE and RAMA cropland data (in earlier versions, including wood harvest). For the 2000s Jain et al. (2013) found an uncertainty of ±0.21 Pg C a$^{-1}$ when

quantifying $E_{LUC}$ with HYDE and RAMA datasets with both a coupled C-N DGVM and the book-keeping approach from Houghton (2008). Our uncertainty of $E_{LUC}$ due to the choice of LUC reconstruction was higher in the 1990s, where $E_{LUC}$ derived using HYDE data was significantly higher due to a strong increase in pastures (Fig. 2a, Table S1). The uncertainty in cumulative $E_{LUC}$ accumulated to ±14 Pg C for 1750-2007. For vegetation and soil carbon, the uncertainty introduced due to the choice of LUC reconstruction was ±11 and ±37 Pg C,

respectively, in 1998-2007, translating into a change of 3 % of the average size of both vegetation and soil C stocks (Table **3**, see Fig. S3 for regional differences for entire simulation period). These uncertainties for vegetation C stocks are higher compared to the ones found by Arora and Boer (2010), who used only two of the LUC models applied here, and about the same size for C stocks in soils. This implies non-linear interactions between DGVM structure and LUC dataset. We would expect uncertainty in C stocks and fluxes to increase, at

least during the pre-1900 period, if LUC reconstructions applying a non-linear development of per capita land use were also considered, such as the KK10 dataset does for the period 8 000 BC to AD 1850 (Kaplan et al., 2010).

For Europe, the uncertainty in $E_{LUC}$ is also large with ±37, ±33 and ±20 Tg C a$^{-1}$ for the 1980s, 1990s and 2000s, which are between 30 % and 72 % of the average flux (Table **4**). Differences result mainly from a disagreement

in the amount of pastures between the LUC reconstructions, where the comparison is impaired by different definitions of the pasture class (see Methods). This highlights the problem of fundamentally different structures and assumptions between LUC models and DGVMs, recently being identified as a major uncertainty in model estimates of $E_{LUC}$ (Pongratz et al., 2014). Although forests, natural grasslands and pastures can show similar gross primary productivity (GPP), they significantly vary in their C sequestration potential in vegetation and

soils also depending on their location within Europe (Ciais et al., 2008; Schulze et al., 2010). For instance larger pasture areas in HILDA in southern Europe compared to LUH1 lead to an increase in vegetation C under LUC, whereas a decrease was simulated with LUH1 (Fig. S4). The productivity and carbon dynamics of croplands in LPJ-GUESS is mainly governed by the crop selection, the bioclimatic conditions of the land where the crop is planted and the degree of fertilization and irrigation (for productivity of croplands under different degrees of

fertilization see also Ciais et al., 2010; Schulze et al., 2010). For instance in Poland, high cropland fractions in LUH1 that were only marginally fertilized compared to croplands in western Europe, decreased NPP under LUC, but increased vegetation and soil C (Fig. S4). Differences between the LUC reconstructions and therefore uncertainties in C stocks and fluxes converge during the first half of the 19[th] century (maps not shown). Vegetation C stocks derived with HILDA are lower than estimates of Fuchs et al. (Fuchs et al., 2015a and

personal communication) using the same datasets (minor difference in net data, see methods) and a bookkeeping method, but are within the uncertainty spanned by using HILDA and LUH1 net LUC datasets (Table S2).

It is important to consider that the relative uncertainties in LUC transitions between datasets are not constant through time, although the absolute uncertainties remain remarkably constant over the simulation period. The relative deviation of pasture area for the three global datasets was about ±29 % before the 1850s, decreasing to
about ±9 % in the 2000s (Fig. S5c). Global cropland areas had a deviation of about ±11 % in the 1700, decreasing to below ±2 % in the 2000s (Fig. S5c). For Europe the agreement of the two LUC reconstructions is high for croplands (average deviation ±2 % for 1900-2010) but lower for pastures and natural areas (1900-2010 on average ±36 % for pastures and ±21 % for natural areas) with the deviation increasing until 2010 for pastures (Fig. S5d). The general agreement on the fractional coverage of natural land, pastures and croplands is higher for
the periods after 1960, when FAO statistical data and later improved data from satellites became available (see also Houghton, 2010; Verburg et al., 2011). Before this period, the extrapolation of historical LUC information was very much dependent on the applied model algorithms in combination with census data. LUC reconstructions also differ in the resolution of past LUC changes, providing annual time steps (RAMA for entire historical period, LUH1, HYDE, LUH2 after 2000) or originating from decadal aggregations (HILDA for entire
historical period, LUH1, HYDE, LUH2 until 2000). Such methodological discrepancies and artifacts from the LUC modeling significantly overlay the observed trends in the LUC reconstructions (compare Fig. 1c, d) and are included in simulated C stocks and fluxes (Fig. 2, Fig. 3). It is noted that LUH1, HYDE and LUH2 net LUC data have LUC features in common and cannot be regarded as completely independent datasets, as LUH data are generally built based on HYDE input data. Here, two subsequent (HYDE as HYDE3.1.1 and LUH2 as based on
HYDE3.2) and one intermediate (LUH1 as based on an early version of HYDE3.2) developments of LUH/HYDE were used (see Methods), however, no versions directly building up on each other were considered (e.g. dataset 1 being a version of HYDE that was used as direct basis for a version of LUH that was considered as dataset 2).

### 4.2.  Uncertainties in carbon stocks and fluxes due to accounting for gross land transitions

The consideration of detailed gross land conversions in our simulations increased the effects of LUC on carbon storage and fluxes due to larger areas transitioning between land-use types (Fig. 1c, d). The increase of about 16 % in average net annual $E_{LUC}$ and 15 % in cumulative $E_{LUC}$ (Table **3**, change due to gross relative to net, 1750-2014) compared well with previous estimates (Table S3). The effect of shifting cultivation on cumulative $E_{LUC}$ was quantified by Olofsson and Hickler (2008) by using the LPJ model (Sitch et al., 2003), which has
similarities in the way plant and soil physiological processes are calculated to the model used here, but a simpler representation of vegetation dynamics and croplands and no C-N dynamics. They found an increase in cumulative $E_{LUC}$ by 28 % and 29 % for 1700-1990 and 1850-1990, whilst Stocker et al. (2014), using a model with coupled C-N dynamics on 1° x 1° spatial resolution, reported an increase by 15 % (despite the coarser spatial resolution acting to accentuate the gross-net differences). Using a bookkeeping model, Hansis et al.
(2015) found an enhancement of $E_{LUC}$ by 22-24% over the period 1500-2012, at a 0.5° x 0.5° spatial resolution. The combined effect of shifting cultivation and wood harvest on cumulative $E_{LUC}$ was summarized by Houghton et al. (2012) as an increase by 25–35 %, and Wilkenskjeld et al. (2014) found an increase in cumulative $E_{LUC}$ of 61 % (51 % without the effect of wood harvest). Shevliakova et al. (2013) provide an estimate of $E_{LUC}$ under gross transitions including wood harvest for the period 1860-2005 using a combination of modeled C fluxes and
a bookkeeping method to derive $E_{LUC}$ that is fairly close to the value calculated in this study. For total land C stocks, Shevliakova et al. (2009) reported an increase in $E_{LUC}$ of 49 % due to shifting cultivation and wood harvest, and concluded from this that the effect on land carbon losses was comparable in magnitude to the effect of cropland and pasture expansion. In our study we quantified an increase in $E_{LUC}$ of 39 % total C globally due to shifting cultivation alone (Table S3). Of these studies, only the model of Stocker et al. (2014) (Table S3),
accounts for C-N interactions. C-N interactions have previously been found to enhance LUC emissions (Jain et al., 2013). In addition, all these studies differ from our study in the DGVM used, LUC data sets and climate model data applied, and the process representations in the models. All studies except Olofsson and Hickler (2008) applied spatial resolutions coarser than the 0.5° applied here (1° or ~2°, see Table S3).

In our global experiment, the only contribution of gross land changes came from shifting cultivation in certain tropical areas. These gross changes were implemented in the LUH1 dataset based on assumed spatial extension and transition rates, and were reflected in significantly increased rates of deforestation and reforestation in gross simulations (Fig. 4a). As would be expected, removing forest material from the system through harvest or burning of cleared vegetation, instead of it entering the soil pool through litter decomposition, reduces soil C content. However, the soil C losses are much less marked than those in vegetation, perhaps reflecting a dominance of vegetation carbon turnover by leaves and fine roots in LPJ-GUESS, which are inputs of C to the litter pool which are less affected by harvesting of vegetation on multi-annual rotation periods than woody inputs.

In comparison to the global gross land transitions, which accounted only for shifting cultivation with uniform assumptions regarding cultivation cycles, the European gross dataset accounted for irregular land-use and land-cover changes based on regionally available empirical evidence (national statistics and maps). $E_{LUC}$ when accounting for gross land changes was about 53 % higher in the first simulation decade and converged to minor differences from about 1980 on (Fig. 3a). Since LUC in Europe developed from being a source of C in the beginning of the 1900s to being a sink after about 1960, the small difference in C stocks and fluxes as a result of accounting for gross transitions, in addition to net land changes, delayed the year in which cumulated $E_{LUC}$ switched from a source to a sink. Thus the sink capacity of cumulative $E_{LUC}$ in the 2000s was reduced, as were increases in vegetation and soil C stocks (Fig. 3b, e, f). To our knowledge, no previous studies are available in which $E_{LUC}$ for Europe was derived under net and gross LUC. The effect of accounting for gross land transitions on vegetation C was negative in our simulations, with vegetation C under gross LUC about 2 % lower than under net LUC, because the increased number of re-growing stands under higher land transitions lowered mean forest canopy closure, which also lowered NPP and, ultimately, soil C (see Table **4**). In contrast Fuchs et al. (Fuchs et al., 2015a and unpublished results), using a bookkeeping method, derived about 1 % higher vegetation C stocks under gross LUC for the same area (Table S2). Discrepancies result from major methodological differences between the bookkeeping and process-based approach, and also from Fuchs et al. not accounting for C stocks in croplands and pastures.

Gross LUC transitions in Europe over the entire simulation period were dominated by conversions between pastures and croplands, i.e. cropland expansion into pasture areas and abandonment of croplands and their conversion into pastures (Fig. 4b), that were direct adjustments to market demands and changes in land-use related policies. Apart from this, LUC in Europe was characterized by abandonment of agricultural land and reforestation peaking in the 1970s. Reforestation of European grasslands was reported to entail a reduction in soil C stocks and an increase in vegetation C (Schulze et al., 2010, Fig. 3e, f), and therefore a positive $E_{LUC}$ (Fig. 3a). After a first period of regrowth, the additional tree biomass and increased litter inputs to soils balanced soil C losses, so that vegetation and soil stocks increased in the second half of the 20th century, because wood harvest was lower than growth (e.g. Ciais et al., 2008), thereby contributing to the LUC sink capacity. Because land abandonment and reforestation are one-directional LUC changes which are represented in the same way in net and gross HILDA data (see Fuchs et al., 2015b), this did not lead to major differences between net and gross simulations. It should be noted that with its current four LUC classes, the LPJ-GUESS model was not able to make full use of the HILDA LUC dataset, as not all HILDA land-cover classes were represented, e.g. urbanization (urban areas were assigned to the barren LUC class, see methods), causing ~18 % of gross land-use changes between 1900-2010 (Fuchs et al., 2015b).

### 4.3. Uncertainties in the modeling approach

The effects and uncertainties discussed above must be considered in relation to other uncertainties arising in the modeling process (e.g. different model implementations in respect to representation of LUC and changes therein, treatment of environmental change). A meta-analysis by Houghton et al. (2012) estimated the uncertainty in $E_{LUC}$ arising from the applied modeling approach and method to be in the range of ±0.2 Pg C a$^{-1}$, and that due to data-related uncertainty and incomplete process understanding to be in the range of ±0.5 Pg C a$^{-1}$. The complex linkages between the contributing factors, however, made it difficult to attribute uncertainties to their sources (see also Jain et al., 2013). Consideration of C-N interaction in vegetation and soils, as was done in this study, is important when studying the effects of environmental drivers such as LUC on carbon emissions, however, with

the exception of a very few models (e.g. Jain et al., 2013; Smith et al., 2014; Xu-Ri and Prentice, 2008; Zaehle and Friend, 2010), most models do not represent N cycling. We did not quantify the effect of C-N interactions in this study, but we note that our estimates of cumulative $E_{LUC}$ from 1850 to 2005 with net HYDE (Table S3) are about 2 % lower than those of Pugh et al. (2015) using a version of LPJ-GUESS without C-N interactions and the same LU data. This is in opposition to the findings of Jain et al. (2013) who found globally about 40 % higher $E_{LUC}$ when accounting for N dynamics and N limitation.

Implementation of gross transitions in a DGVM framework are subject to considerable uncertainty. By varying the minimum age upon which a regrowing natural stand becomes eligible for clearance again between 5 and 30 years (recalling that natural stands are removed in order of age when natural vegetation is reduced, see Methods), cumulative $E_{LUC}$ was found to differ by ±20 Pg C (10 %) (1900-2014, LUH1 only, results not shown). Hence, choosing a too-young clearance age would lead to considerable underestimates of $E_{LUC}$ which highlights an important, and hitherto unremarked upon, implementation uncertainty for including gross transitions in DGVM simulations. In the same way, other land-use and land-cover change-related processes (e.g. fate of harvested wood, residue management, occurrence of disturbances, including fire, etc.) might differ under repeated land transitions such as shifting cultivation, and realistic representation of these interactions in process-based models remains a subject for further research. Wood harvest mainly in extra-tropical regions was assessed to account for an increase in $E_{LUC}$ of 0.2-0.3 Pg C $a^{-1}$ (Houghton et al., 2012). We did not account for wood harvest in this study as the uncertainty in the actual spatial pattern of wood harvest in combination with the ways wood harvest is done in practice over the globe (clear cut, selective harvesting of specific age classes or a mixture of both) introduces many possibilities as to how this process can be implemented in DGVMs. In a model such as LPJ-GUESS, where forest ecosystem and wood parameters vary significantly over tree age classes, this would result in a wide span of possible solutions depending on the parameters used for implementation of wood harvest that would be better addressed in a dedicated sensitivity study investigating a variety of possible implementations, rather than with a single representation.

Simulation results of biogeochemical cycles with a DGVM such as LPJ-GUESS depend critically on the year when simulations are started. In this study we tested the effect of starting LUH1 simulations in either 1700 or 1900 (Fig. S6). $E_{LUC}$ cumulated over 1950-2014 was 17 % lower and soil C was about 33 Pg C (2 %) lower when simulations were started in 1900, while vegetation C stocks were similar (net LUC, Table S4). This emphasizes the impact on simulation results of soil legacy emissions resulting from previous LUC changes (see, e.g. Gasser and Ciais, 2013, and Sentman et al., 2011 for legacy fluxes, and Pugh et al., 2015 for breakdown of LUC emissions). Hansis et al. (2015) in contrast found a 28% higher $E_{LUC}$ over the period 2008-2012 when legacy emissions predating 2008 were excluded. This resulted from previous LUC transitions generating more uptake from regrowth than loss from soil decomposition during the 2008-2012 period, and further exemplifies the considerable importance of legacy effects for calculation of instantaneous emissions. They also emphasize that the change in soil and vegetation stocks induced by previous LUC can substantially modify $E_{LUC}$ calculations compared to an assumption of equilibrium in the initial stocks. Often such changes in initial conditions tend to lower vegetation C stocks and thus subsequent deforestation emissions. Our 17% lower $E_{LUC}$ excluding legacy emissions accounts for both these effects. To exclude this effect from the analysis of uncertainties due to LUC dataset selection and the effect of accounting for gross LUC transitions, all global simulations were started in 1700. For Europe, where HILDA reconstructions were not available before 1900, simulations were started in 1900 for both HILDA and LUH1 to ensure comparability.

## 5. Conclusions

Global and European carbon stocks and fluxes and the effects of changes in LUC were shown to be subject to significant uncertainties resulting from the choice of historical LUC reconstruction. In our global simulations, HYDE/ RAMA and LUH1/LUH2 data often lead to similar results in ecosystem C stocks and fluxes. Therefore LUH1 and LUH2 as the more recent developments under the four considered reconstructions (both based on HYDE version 3.2, however LUH1 on an intermediate version and LUH2 on the final version) differ more from older developments than these from each other (see methods for model versions used in this study). For Europe,

variables predicted based on the newly available HILDA dataset were similar to those resulting from using LUH1 for Europe, however LUH1 predicted larger changes under LUC. Differences in the effects of both global and European LUC on C stocks and fluxes were found to be mainly based on the total area and spatial distribution of pastures in the datasets, however noting that the area of pastures is impaired by different classifications used by the LUC models. To account for the uncertainty arising from different reconstructions of historical LUC in the dynamic modeling of C stocks and fluxes and to provide realistic estimates of this uncertainty for the land-use C flux, the consideration of multiple LUC reconstructions exploring the full range of reasonable assumptions is needed, as well as efforts to narrow the uncertainty in constructions of historical land use. Multiple LUC reconstructions were calculated by Hurtt et al. (2011), but the consequences of uncertainty in land-use transitions are not routinely explored by the carbon cycle community. This goes along with the reduction of uncertainties in the implementation of these datasets and different forms of LUC in DGVMs which has recently been the focus of discussion (Pongratz et al., 2014; Pugh et al., 2015; Stocker and Joos, 2015).

The results herein showed that considering gross land conversions significantly increased the effect of LUC change on C stocks and fluxes. Most noticeably the land-use C flux was enhanced by about 15 % of carbon released in addition to when only accounting for net land changes (cumulated 1750-2014), primarily resulting from a reduction in vegetation C storage. Note that for DGVMs operating at a lower spatial resolution than the 0.5° x 0.5° used here, the underestimation of $E_{LUC}$ would be even larger, as shown by Wilkenskjeld et al. (2014). Given the large percentage enhancements in $E_{LUC}$ found by considering gross transitions, this should be the preferred method whenever possible.

Implementation of gross land transitions, however, poses technical and parameterization challenges to the process-based models. It also relies on extensive information on historical land-use transitions, which is largely lacking; at present, only a few LUC models are able to represent gross land changes on larger spatial scales, providing a limited basis to characterize the uncertainty. The LUH datasets are the only global scale reconstructions representing gross land changes by explicitly implementing shifting cultivation in certain tropical areas with assuming a fixed period of 15 years for which land is cultivated before abandonment (for LUH1). For Europe, the HILDA data set is the first reconstruction representing gross land transitions which are based on actual LUC inventory data and complementary model assumptions. The reconstruction of detailed regional sub-grid land transitions, and possibly more realistic patterns of shifting cultivation today, is restricted by the lack of reliable information on continental and global-scale historical land transitions. New datasets based on archived LUC data and remote sensing sources are currently becoming available with high spatial resolution for global to continental scale (Chen et al., 2015; European Environment Agency, 2014; Wang et al., 2015) and on national to regional level (Homer et al., 2015; MOFOR, 2016; RCMRD, 2016; Roy et al., 2015; TerraClass, 2016). New promising efforts also provide LUC change data globally derived from remote sensing with 250m spatial resolution (Wang et al., 2015) and with 30m spatial resolution (Chen et al., 2015). In the coming years, new high resolution LUC datasets can be expected from the Landsat archives (http://earthexplorer.usgs.gov) and the new Sentinel missions (Aschbacher and Milagro-Pérez, 2012). These will contribute to close the information gap and with this improve the calibration of LUC models to represent the underlying processes, reduce the uncertainty in ecosystem functions such as the present-day land-use flux, and provide enhanced information, for, e.g., the assessment of ecosystem services and biodiversity indicators in the future.

**Acknowledgements**

This work was funded by the European Commission's 7th Framework Program under Grant Agreement numbers 308393 (OPERAs) and 603542 (LUC4C). This work was supported, in part, by the German Federal Ministry of Education and Research (BMBF), through the Helmholtz Association and its research program ATMO. The authors thank Stefan Olin from Lund University for advice on C-N cycling in crops. This is paper number 23 of the Birmingham Institute of Forest Research.

**Author contributions**

AB, TP and AA conceived and designed the experiments. ML realized the technical implementation of gross land-use changes in LPJ-GUESS. PA processed LUH2 LUC data. AB carried out the model simulations and led the data analysis, with contributions from all authors. RF contributed the HILDA land-use data. AB led the writing of the manuscript with contributions from all authors.

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

Table 1. Raw characteristics of LUC datasets used in this study[a].

| Land-use model | | time period | | time steps | Representation of LUC transitions | Spatial coverage | Original spatial resolution | Land cover classes |
|---|---|---|---|---|---|---|---|---|
| | Model version and reference | from | to | | | | | |
| LUH1 | 1500-2005: LUH1 (Hurtt et al., 2011), extension until 2014: Le Quéré et al. (2015b)[b] | 1500 | 2014[c] | annual | net/gross | global | 0.5° | cropland, pasture, primary natural vegetation, secondary natural vegetation, urban |
| RAMA | 1700-1992: Ramankutty and Foley (1999), extension until 2007: Ramankutty (2012) | 1700 | 2007 | annual | net | global | 0.5° | cropland, pasture, primary natural vegetation, secondary natural vegetation, urban |
| HYDE | 10 000 BC to AD 2000: HYDE3.1.1 (Klein Goldewijk et al., 2010, 2011), extension until 2005: see Klein Goldewijk et al.( 2015), extension until 2013: see Le Quéré et al. (2015a) | 10.000 BC | 2013 | annual[d] | net | global | 5' | cropland, pasture, natural vegetation[e] |
| LUH2 | 850-2015: LUH2 v2 (release 14 Oct 16), (Hurtt et al., 2016) | 850 | 2015 | annual | net[f] | global | 0.25° | C3/C4 annual/perennial crops, C3 N-fixing crops, managed pasture, rangeland, primary natural vegetation, secondary natural vegetation, forest, non-forest, urban |
| HILDA | HILDA v2.0 (Fuchs et al., 2015b) | 1900 | 2010 | decadal | net[g]/gross | EU27[h] plus Switzerland | 1 km | cropland, grassland (incl. managed pastures and shrubland), forest, settlements, water, other land (glaciers, sparsely vegetated areas, beaches and water bodies) |

[a] Note that these datasets are preprocessed into more consistent characteristic sets before carrying out simulations - see methods and Table 2. [b] Note that this version of LUH1 is based on an early version of HYDE 3.2, which is different from HYDE version 3.1.1 as used below, see methods. [c] End date of historical land use data set. [d] Decadal data until 2000. [e] Natural vegetation is calculated as a remainder. [f] Only the net dataset was used in this study (see methods). [g] The HILDA net dataset used in this study was derived from the gross dataset (see methods). [h] European Union 2007-2013.

Table 2. Overview of LPJ-GUESS simulations carried out as part of this study.

| Abbreviation | First year | Last year | Representation of LUC transitions | Spatial coverage |
|---|---|---|---|---|
| LUH1 | 1700 | 2014 | gross | global |
| | 1700 | 2014 | net | global |
| | 1700 | 2014 | LUC fixed to 1700 | global |
| RAMA | 1700 | 2007 | net | global |
| | 1700 | 2007 | LUC fixed to 1700 | global |
| HYDE | 1700 | 2013 | net | global |
| | 1700 | 2013 | LUC fixed to 1700 | global |
| LUH2 | 1700 | 2015 | net | global |
| | 1700 | 2015 | LUC fixed to 1700 | global |
| HILDA | 1900 | 2010 | gross | EU27+CH |
| | 1900 | 2010 | net | EU27+CH |
| | 1900 | 2010 | LUC fixed to 1900 | EU27+CH |
| LUH1 | 1900 | 2014 | net | EU27+CH |
| | 1900 | 2014 | LUC fixed to 1900 | EU27+CH |

Table 3. Changes in C stocks and fluxes in four global reconstructions of net and gross LUC changes. Land use change flux ($E_{LUC}$) and cumulative land use flux $E_{LUC}$, Net primary productivity (NPP), C stocks in vegetation and soils. Values are always given as 10-year averages (except LUC areas and cumulative flux).

| | Averaging period | Calculation | Unit | LUH1 net | RAMA net | HYDE net | LUH2 net | Average and uncertainty for 4 net LUC models | LUH1 gross | difference LUH1 (gross-net) |
|---|---|---|---|---|---|---|---|---|---|---|
| **Natural area** | 1700 | $A_{nat}$ | $10^6$ km$^2$ | 122.48 | 124.35 | 126.54 | 123.23 | 124.15 ± 1.77 | 122.48 | 0 |
| **Natural area** | 2007 | $A_{nat}$ | $10^6$ km$^2$ | 85.39 | 90.31 | 84.97 | 85.01 | 86.42 ± 2.60 | 85.39 | 0 |
| **Pasture area** | 1700 | $A_{pas}$ | $10^6$ km$^2$ | 7.45 | 4.89 | 3.32 | 6.61 | 5.57 ± 1.84 | 7.45 | 0 |
| **Pasture area** | 2007 | $A_{pas}$ | $10^6$ km$^2$ | 32.38 | 27.04 | 32.81 | 32.73 | 31.24 ± 2.81 | 32.38 | 0 |
| **Cropland area** | 1700 | $A_{crop}$ | $10^6$ km$^2$ | 2.80 | 3.49 | 2.86 | 2.89 | 3.01 ± 0.32 | 2.80 | 0 |
| **Cropland area** | 2007 | $A_{crop}$ | $10^6$ km$^2$ | 14.96 | 15.38 | 14.95 | 14.99 | 15.07 ± 0.21 | 14.96 | 0 |
| **Change in natural area** | 1700-2007 | $A_{nat}$ | $10^6$ km$^2$ | -37.09 | -34.04 | -41.57 | -38.22 | -37.73 ± 3.11 | -37.09 | 0 |
| **Change in pasture area** | 1700-2007 | $A_{pas}$ | $10^6$ km$^2$ | +24.93 | +22.15 | +29.49 | +26.12 | +25.67 ± 3.04 | +24.93 | 0 |
| **Change in cropland area** | 1700-2007 | $A_{crop}$ | $10^6$ km$^2$ | +12.16 | +11.89 | +12.09 | +12.10 | 12.06 ± 0.12 | +12.16 | 0 |
| **Total area under transition** | 1700-2007 | $\Sigma A_{trans}$ | $10^6$ km$^2$ | 51.92 | 78.59 | 64.62 | 57.28 | 63.10 ± 11.56 | 243.80 | +191.88 |
| **Change in area under transition** | 1850-1960 | delta$A_{trans}$ | km$^2$ a$^{-1}$ | +5137 | +7259 | +6703 | +5177 | 6069 ± 1077 | +5549 | +412 |
| **$E_{LUC}$** | 1750-2007 | $E_{LUC\ Net/Gross}$ | Pg C a$^{-1}$ | 0.81 | 0.87 | 0.89 | 0.77 | 0.84 ± 0.05 | 0.94 | + 0.13 |
| | 1750-2014 | $E_{LUC\ Net/Gross}$ | Pg C a$^{-1}$ | 0.84 | - | - | 0.80 | - | 0.96 | + 0.12 |
| | 1980-1989 | $E_{LUC\ Net/Gross}$ | Pg C a$^{-1}$ | 1.10 | 1.40 | 1.55 | 1.31 | 1.34 ± 0.19 | 1.28 | +0.18 |
| | 1990-1999 | $E_{LUC\ Net/Gross}$ | Pg C a$^{-1}$ | 1.18 | 1.57 | 2.65 | 1.36 | 1.69 ± 0.66 | 1.41 | +0.23 |
| | 1998-2007 | $E_{LUC\ Net/Gross}$ | Pg C a$^{-1}$ | 1.17 | 2.00 | 2.06 | 1.26 | 1.62 ± 0.47 | 1.38 | + 0.20 |
| | 2005-2014 | $E_{LUC\ Net/Gross}$ | Pg C a$^{-1}$ | 1.50 | - | - | 1.67 | - | 1.64 | + 0.14 |
| **Cumulative $E_{LUC}$** | 1750-2007 | $\Sigma E_{LUC\ Net/Gross}$ | Pg C | 210.02 | 225.18 | 228.95 | 199.12 | 215.82 ± 13.82 | 242.04 | + 32.02 |
| | 1750-2014 | $\Sigma E_{LUC\ Net/Gross}$ | Pg C | 222.29 | - | - | 212.70 | - | 255.27 | + 32.98 |
| **NPP** | 1700-1709 | $NPP_{Net/Gross}$ | Pg C a$^{-1}$ | 50.18 | 52.10 | 52.21 | 50.43 | 51.23 ± 1.07 | 50.04 | -0.14 |
| | 1998-2007 | $NPP_{Net/Gross}$ | Pg C a$^{-1}$ | 58.90 | 59.79 | 60.85 | 59.14 | 59.67 ± 0.87 | 57.49 | -1.42 |
| | 2005-2014 | $NPP_{Net/Gross}$ | Pg C a$^{-1}$ | 59.95 | - | - | 60.18 | - | 58.46 | -1.49 |
| **Change in NPP due to LUC** | 1998-2007 | $NPP_{Net/Gross}$-$NPP_{Ref}$ | Pg C a$^{-1}$ | -1.92 | -3.32 | -2.44 | -1.96 | -2.41 ± 0.65 | -3.33 | -1.42 |
| | 2005-2014 | $NPP_{Net/Gross}$-$NPP_{Ref}$ | Pg C a$^{-1}$ | -2.26 | - | - | -2.30 | - | -3.75 | -1.49 |
| **Vegetation C** | 1700-2014 | $VegC_{Net/Gross}$ | Pg C | 435 | - | - | 444 | - | 419 | -16 |
| | 1700-1709 | $VegC_{Net/Gross}$ | Pg C | 464.18 | 495.70 | 497.19 | 467.59 | 481. 17 ± 17.71 | 461.35 | -2.83 |
| | 1998-2007 | $VegC_{Net/Gross}$ | Pg C | 414.62 | 439.10 | 435.22 | 424.53 | 428.37 ± 11.04 | 380.43 | -34.20 |
| | 2005-2014 | $VegC_{Net/Gross}$ | Pg C | 421.48 | - | - | 431.41 | - | 386.64 | -34.84 |
| **Change in vegetation C due to LUC** | 1960-1969 | $VegC_{Net/Gross}$-$VegC_{Ref}$ | Pg C | -109.49 | -114.12 | -108.21 | -98.56 | -107.60 ± 6.54 | -137.06 | -27.58 |
| | 1998-2007 | $VegC_{Net/Gross}$-$VegC_{Ref}$ | Pg C | -140.36 | -153.41 | -159.42 | -134.36 | -146.89 ± 11.54 | -174.56 | -34.20 |
| | 2005-2014 | $VegC_{Net/Gross}$- $VegC_{Ref}$ | Pg C | -148.43 | - | - | -142.58 | - | -183.27 | -34.27 |
| **Soil C** | 1700-2014 | $SoilC_{Net/Gross}$ | PgC | 1 425.85 | - | - | 1 438.39 | - | 1 420.81 | -5.04 |
| | 1700-1709 | $SoilC_{Net/Gross}$ | PgC | 1 445.96 | 1 511.65 | 1 516.06 | 1 453.68 | 1481.84 ± 37.15 | 1 445.58 | -0.38 |
| | 1998-2007 | $SoilC_{Net/Gross}$ | Pg C | 1 404.20 | 1 471.83 | 1 478.08 | 1 419.90 | 1 443.50 ± 36.97 | 1 393.43 | -10.77 |
| | 2005-2014 | $SoilC_{Net/Gross}$ | Pg C | 1 406.78 | - | - | 1 421.70 | - | 1 395.56 | -11.22 |
| **Change in soil C due to LUC** | 1998-2007 | $SoilC_{Net/Gross}$-$SoilC_{Ref}$ | Pg C | -75.80 | -75.04 | -73.54 | -67.95 | -73.08 ± 3.55 | -86.57 | -10.77 |
| | 2005-2014 | $SoilC_{Net/Gross}$-$SoilC_{Ref}$ | Pg C | -77.74 | - | - | -70.59 | - | -88.96 | -11.22 |

Table 4. Changes in C stocks and fluxes in reconstructions of net and gross LUC changes for Europe (EU27+CH). Values are always given as 10-year averages (except LUC
areas and cumulative flux).

| | Averaging period | Calculation | Unit | HILDA (net) | LUH1 (net) | Average and uncertainty[a] for the LUC models | HILDA (gross) | difference HILDA (gross-net) |
|---|---|---|---|---|---|---|---|---|
| **Natural area** | 1900 | $A_{nat}$ | $10^6$ km$^2$ | 1.49 | 2.16 | 1.83 ± 0.47 | 1.49 | 0 |
| **Natural area** | 2010 | $A_{nat}$ | $10^6$ km$^2$ | 2.06 | 2.79 | 2.43 ± 0.52 | 2.06 | 0 |
| **Pasture area** | 1900 | $A_{pas}$ | $10^6$ km$^2$ | 1.62 | 0.97 | 1.30 ± 0.46 | 1.62 | 0 |
| **Pasture area** | 2010 | $A_{pas}$ | $10^6$ km$^2$ | 1.35 | 0.68 | 1.02 ± 0.47 | 1.35 | 0 |
| **Cropland area** | 1900 | $A_{crop}$ | $10^6$ km$^2$ | 1.59 | 1.57 | 1.58 ± 0.01 | 1.59 | 0 |
| **Cropland area** | 2010 | $A_{crop}$ | $10^6$ km$^2$ | 1.29 | 1.23 | 1.26 ± 0.04 | 1.29 | 0 |
| **Total change in natural area** | 1900-2010 | $A_{nat}$ | $10^6$ km$^2$ | +0.57 | +0.63 | +0.60 ± 0.04 | +0.57 | 0 |
| **Total change in pasture area** | 1900-2010 | $A_{pas}$ | $10^6$ km$^2$ | -0.29 | -0.34 | -0.28 ± 0.01 | -0.29 | 0 |
| **Total change in cropland area** | 1900-2010 | $A_{crop}$ | $10^6$ km$^2$ | -0.28 | -0.29 | -0.32 ± 0.03 | -0.28 | 0 |
| **Total area under transition** | 1900-2010 | $\Sigma A_{trans}$ | $10^6$ km$^2$ | 1.47 | 1.87 | 1.67 ± 0.29 | 2.64 | +1.17 |
| **$E_{LUC}$** | 1900-1909 | $E_{LUC\ Net/Gross}$ | Tg C a$^{-1}$ | 19 | 38 | 29 ± 13 | 29 | +9 |
| | 1980-1989 | $E_{LUC\ Net/Gross}$ | Tg C a$^{-1}$ | -25 | -78 | -51 ± 37 | -26 | -1 |
| | 1990-1999 | $E_{LUC\ Net/Gross}$ | Tg C a$^{-1}$ | -38 | -84 | -61 ± 33 | -38 | 0 |
| | 2001-2010 | $E_{LUC\ Net/Gross}$ | Tg C a$^{-1}$ | -52 | -80 | -66 ± 20 | -51 | +1 |
| **cum($E_{LUC}$)** | 1951-1960 | $\Sigma E_{LUC\ Net/Gross}$ | Tg C | 586 | 1338 | 962 ± 532 | 915 | +329 |
| | 2010 | $\Sigma E_{LUC\ Net/Gross}$ | Tg C | -936 | -1 890 | 674 ± 48 | -531 | +406 |
| **NPP** | 1900-1909 | $NPP_{Net/Gross}$ | Tg C a$^{-1}$ | 1 464 | 1 517 | 1 490 ± 38 | 1462 | -2 |
| | 2001-2010 | $NPP_{Net/Gross}$ | Tg C a$^{-1}$ | 2 261 | 2 361 | 2 311 ± 71 | 2243 | -18 |
| **Change in NPP due to LUC** | 1900-2010 | $NPP_{Net/Gross}$-$NPP_{Ref}$ | Tg C a$^{-1}$ | -30 | -10 | -20 ± 14 | -44 | -14 |
| | 2001-2010 | $NPP_{Net/Gross}$-$NPP_{Ref}$ | Tg C a$^{-1}$ | -57 | +10 | -23 ±47 | -74 | -18 |
| **VegC** | 1900-1909 | $VegC_{Net/Gross}$ | Tg C | 7 755 | 9 634 | 8 694 ± 1328 | 7 715 | -40 |
| | 2001-2010 | $VegC_{Net/Gross}$ | Tg C | 10 518 | 13 484 | 12 001 ± 2097 | 10 360 | -159 |
| **Change in vegetation C due to LUC** | 1900-2010 | $VegC_{Net/Gross}$- $VegC_{Ref}$ | Tg C | -58 | -234 | -146 ± 125 | -199 | -141 |
| | 2001-2010 | $VegC_{Net/Gross}$-$VegC_{Ref}$ | Tg C | +709 | +1217 | +963 ± 359 | +551 | -159 |
| **Soil C** | 1900-1909 | $SoilC_{Net/Gross}$ | Tg C | 58 786 | 60 672 | 59 729 ± 1 334 | 58 775 | -11 |
| | 2001-2010 | $SoilC_{Net/Gross}$ | Tg C | 60 016 | 62 188 | 59 761 ± 1 536 | 59 761 | -254 |
| **Change in soil C due to LUC** | 1900-2010 | $SoilC_{Net/Gross}$-$SoilC_{Ref}$ | Tg C | -199 | -314 | -256 ± 81 | -368 | -169 |
| | 2001-2010 | $SoilC_{Net/Gross}$-$SoilC_{Ref}$ | Tg C | -29 | +291 | +131 ± 226 | -283 | -254 |

[a] Note that the uncertainty given here is calculated between 2 values.

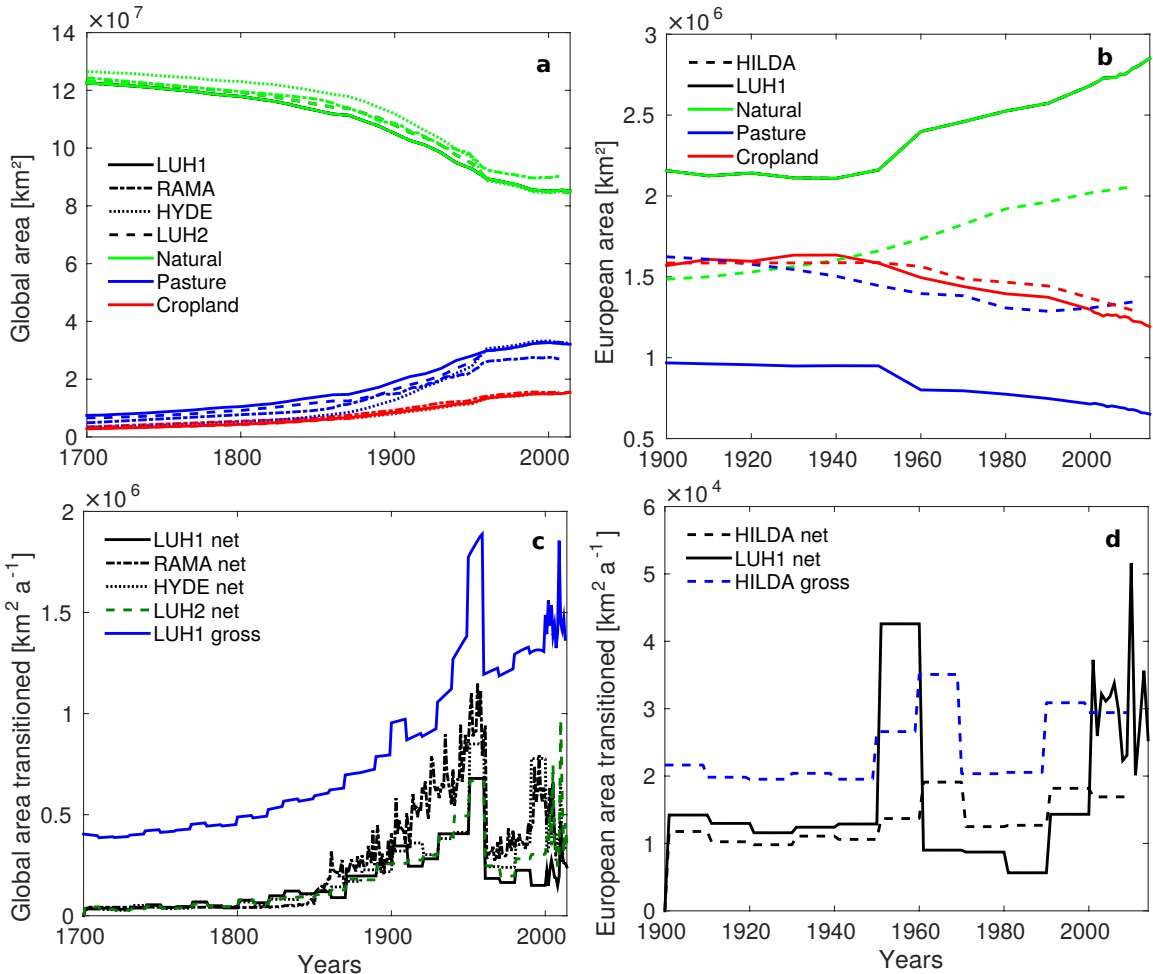

Fig. 1. Land use types and transitions in global and European (EU27+CH) LUC reconstructions. Evolution of
absolute land area of croplands, pastures and natural vegetation (including barren land) in different (a) global
historical land use reconstructions (LUH1: solid line, RAMA: dash-dotted line, HYDE: dotted line, LUH2:
dashed grey line), and (b) European land use reconstructions (HILDA: dashed line, LUH1: solid line). Land area
experiencing gross and net land transitions on global scale (c) and for Europe (d). Note the change to annual
resolution in the LUH1 reconstruction after the year 2000.

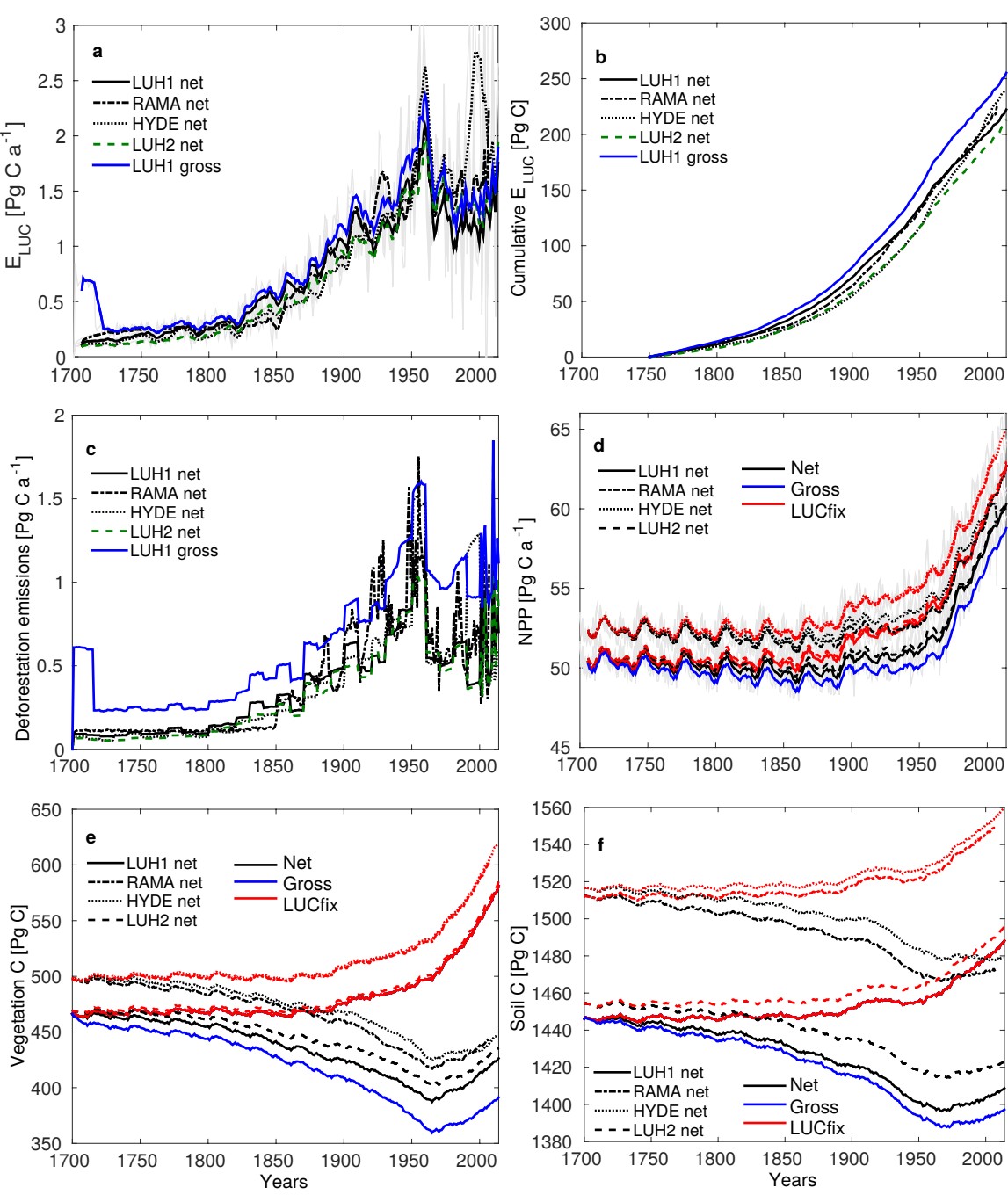

Fig. 2. Effects of different land use representations on global ecosystem C stocks and fluxes: Land use flux (a),
cumulative land use flux (b), deforestation emissions (c), Net primary productivity (NPP) (d), vegetation (e) and
soil carbon stocks (f). Flux values in (a) and (d) are given as 15-yrs averages with original values in the
background. NPP, vegetation and soil C is shown for simulations experiencing net (or gross) LUC and in
addition for simulations where LUC was kept at 1700 levels (see methods, red lines in d, e, f).

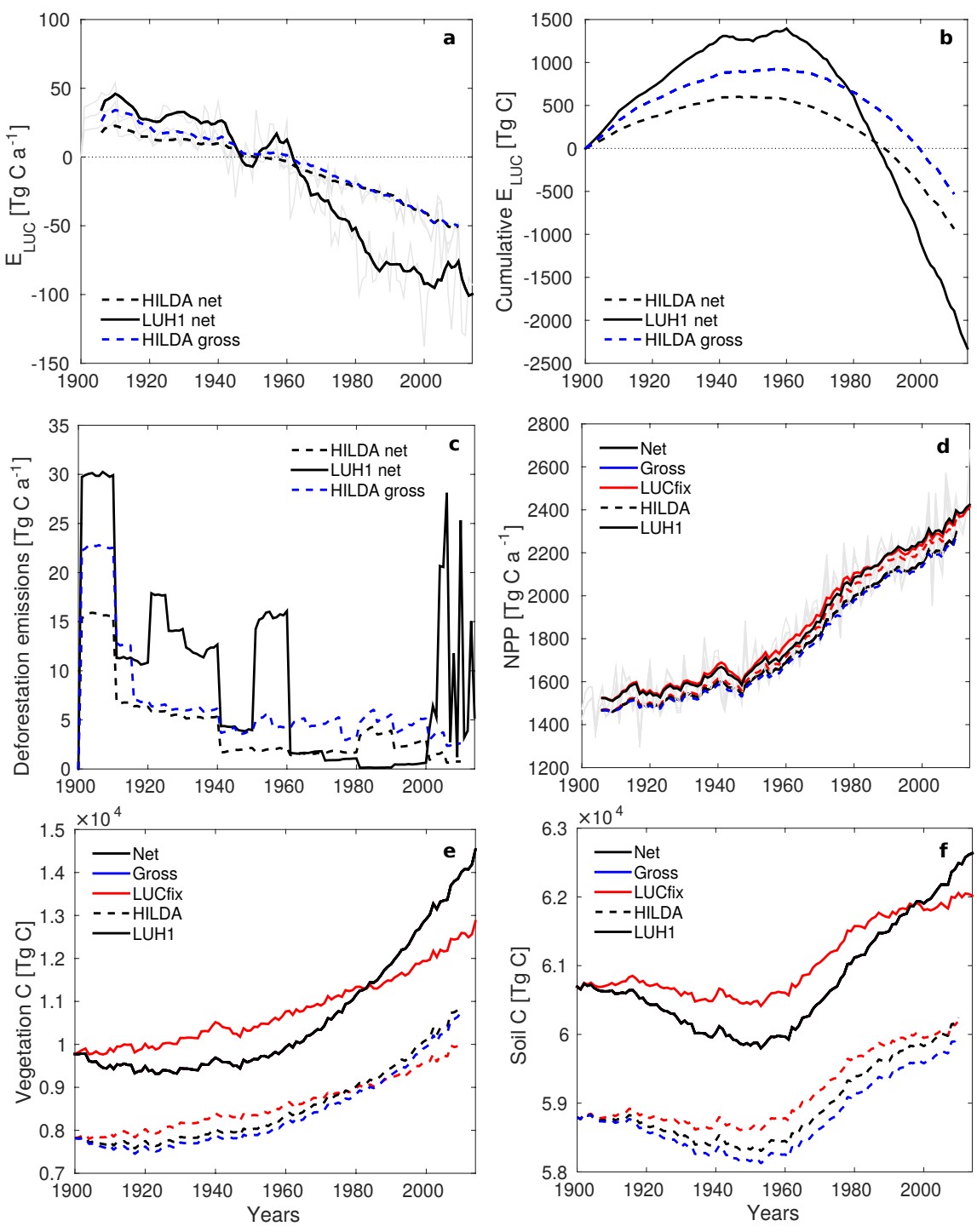

Fig. 3. Effects of different land use representations on ecosystem C stocks and fluxes for Europe (EU27+CH):
Land use flux (a), cumulative land use flux (b), deforestation emissions (c), Net primary productivity (NPP) (d),
vegetation (e) and soil C (f). Flux values in (a) and (d) are given as 15-yrs averages with original values in the
background. NPP, vegetation and soil C is shown for simulations experiencing net (or gross) LUC and in
addition for simulations where LUC was kept at 1700 levels (see methods, red lines in d, e, f).

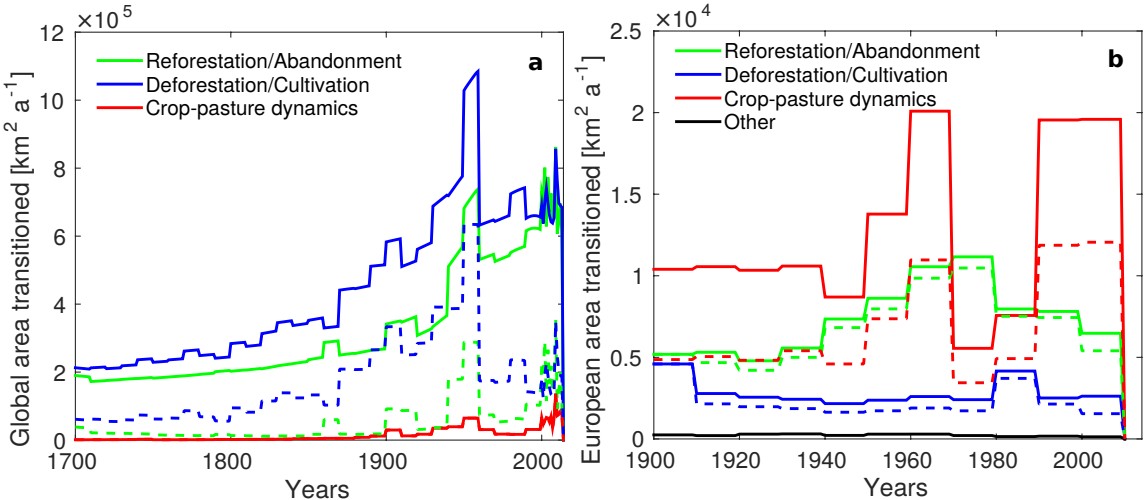

Fig. 4. Annual area transitions for major land change processes for global (a, data from LUH1) and European
(b), EU27+CH, data from HILDA) gross (solid lines) and net (dashed lines) datasets. The class "other" in (b)
includes transitions between natural vegetation and barren land represented in the HILDA dataset.