# Peer review of "Uncertainties in the land use flux resulting from land use change reconstructions and gross land transitions"

_Earth System Dynamics, 2016_

## Referee Comment (RC1) · Anonymous Referee #1 · 22 Jul 2016

Summary: In this paper the authors use a dynamic global vegetation model (DGVM) to calculate the ecosystem carbon stocks and fluxes that result from the use of different land-use change reconstructions. Several historical period land-use datasets are used – three that have a global domain and one that has a European domain. In addition, two of the historical land-use datasets provide gross land-use transitions rather than simply the net land-use transitions in each grid-cell, and the effect of gross vs. net transitions on the carbon stocks and fluxes is investigated along with the uncertainties arising from the choice of land-use dataset used. It is an interesting study because the representation of land-use within DGVMs and climate models is very uncertain, and also very important.

[Figure]

Recommendation: I recommend this paper for publication subject to the following

1. The authors should clarify exactly which version of the HYDE dataset is being used in this study. In Table 1 they cite Klein Goldewijk 2015 but that paper is not listed in the Reference section. They also state that they are using the version of HYDE and the version of LUH used in Le Quere et al. 2015. However those datasets should be identical for global areas of cropland and pasture, although they are apparently not identical in this study (from Figure 1). A statement that the version of HYDE used in this study is not the same as the version of HYDE used as an input to the LUH dataset would be helpful. In addition, if the version of LUH used in this study is indeed the same one used in Le Quere et al. 2015, it would be good to state that this version of the LUH dataset differs from the standard LUH dataset used in most CMIP5 experiments.

2. In the abstract the authors state that the main reason that gross land-use transitions have previously not been included in carbon modeling studies is the lack of detailed information on historical gross land-use changes. However, I would also argue that until recently many carbon models were not able to use even the simple gross land-use changes provided by land-use datasets.

3. Another clarification: the LUH dataset includes shifting cultivation only in some locations within the tropics (based on the map of Butler 1980). There are currently several places in the paper where it is implied that shifting cultivation occurs throughout the tropics.

4. Although the authors use both net and gross land-use transitions in this study, they do not describe how they determine the net transitions for the LUH dataset (which by default provides gross transitions) or the HYDE dataset (which does not provide transitions at all – just land-use states). The calculation of net transitions should not be difficult in either case, but a brief description should be included for completeness in the methods section.

5. The lack of wood harvest is a limitation of the modeling approach used by the authors

and it would be good to include some more discussion of this. Wood harvest is one of the largest land-use transitions in terms of both area and carbon emissions. Although the spatial pattern of wood harvest is uncertain, national data on wood harvest amounts and areas are available. When comparing the effects of including only net vs. gross transitions it is important to consider that wood harvest is a gross transition that is currently not included in this study.

6. The Discussion section begins by stating that a key uncertainty in estimating C stocks and fluxes from land-use stems from the choice of LUC dataset used. I think it would be good to rephrase this opening statement slightly to remind readers that historical reconstructions of land-use are inherently uncertain, and it is not just the choice of LUC dataset that introduces uncertainty. For example, *all* LUC datasets used in this study show a peak in LU transitions around 1950-1960, although there is some evidence that this is likely due to the reconstruction process itself (i.e. the merging of two or more data sources during that time period).

7. In the Conclusion section the authors state that the consideration of multiple LUC reconstructions exploring the full range of reasonable assumptions is needed. This was actually a central component of the paper of Hurtt et al. 2011 in which those authors performed a large sensitivity study by varying all model inputs and decision parameters to explore a range of possible land-use reconstructions.

8. Also in the Conclusion section the authors state that the differences in C stocks and fluxes predicted by the HYDE and LUH datasets is surprising given that they are based on the same data inputs etc. However, it appears that two different (inconsistent) versions of these datasets were used – see comment 1 above.

---

## Referee Comment (RC2) · Anonymous Referee #2 · 30 Sep 2016

The manuscript by Bayer et al compares and contrasts the implementation of different land cover change products on carbon emissions. Four land cover datasets were considered, and the implementation varied by using either net or gross transitions. The LPJ-GUESS model, modified to include pasture/crop management, as well as deforestation and regrowth, was used to estimate carbon fluxes with the different land cover modeling data and approaches. Accounting for sub-grid (gross) land cover conversions had significant effects on carbon stocks and fluxes. The study reinforces the importance of how the implementation of land cover change can significantly alter the estimation of land cover change carbon emissions.

The paper is very well organized and clearly described, there are a few citation problems that can be easily addressed.

Mainly, the manuscript would be strengthened by:

1. Including in the discussion a critique of how wood harvest can be included in the future, and what sorts of feedbacks would be expected on the component flux emissions

2. Is forest age structure included in the model? The authors mention that 'young stands are harvested before old stands'. This implies there is age structure. Also, what is the logic for harvesting the young stands first rather than the old stands? What would be the implications of doing this in reverse?

3. Fire is not discussed in the manuscript – how would the authors plan to include fire feedbacks and the reorganization of forest structure in their sub-grid cell based transitions?

---

## Author Comment (AC1) · 30 Nov 2016

**Response to Interactive comment to anonymous referee #1**

Thank you for the positive feedback and the suggestions to improve the manuscript. We agree that these are very relevant points that, after inclusion, make the manuscript much clearer to follow. Below are the questions and suggestions followed by answers including the changes that were made in the manuscript.

In addition to addressing the reviewers' comments we extended the analysis to including the newly available LUH2 net dataset in the analysis. Since this dataset will be used e.g. in CMIP6 simulations, we expect this dataset to emerge as one that will frequently be used for modeling studies in the future. As this we included it in the analysis. The inclusion of LUH2 in averages and uncertainties in C stocks and fluxes did not change the outcomes of this study. We also include a section on the inclusion of LUH2 as a model that is indeed not completely independent from LUH1 and HYDE in the discussion.

*1. The authors should clarify exactly which version of the HYDE dataset is being used in this study. In Table 1 they cite Klein Goldewijk 2015 but that paper is not listed in the Reference section. They also state that they are using the version of HYDE and the version of LUH used in Le Quere et al. 2015. However those datasets should be identical for global areas of cropland and pasture, although they are apparently not identical in this study (from Figure 1). A statement that the version of HYDE used in this study is not the same as the version of HYDE used as an input to the LUH dataset would be helpful. In addition, if the version of LUH used in this study is indeed the same one used in Le Quere et al. 2015, it would be good to state that this version of the LUH dataset differs from the standard LUH dataset used in most CMIP5 experiments.*

We agree with the reviewer that we were not totally clear on the exact versions of the datasets that were used in the study. We followed up with the developers of the datasets and added information to the text and Table 1 that should provide a clear identification of the datasets that were used. This also includes a statement regarding CMIP5 and CMIP6 simulations. We apologise for omitting Klein Goldewijk (2015) from the reference list and have corrected this.

§2.1, lines 155-157: *The HYDE dataset used here was extended until 2005 (Klein Goldewijk et al., 2015), and later until 2013 in the 2014 global carbon budget analysis (Le Quéré et al., 2015a).*

§2.1, lines 169-178: *The LUH1 dataset was extended until 2014 for the 2015 global carbon budget analysis (Le Quéré et al., 2015b), using an early version of HYDE 3.2 as the basis (now published in final version as Klein Goldewijk, 2016) and following the same methodology as described in Hurtt et al. (2011). The version of LUH1 used in this study is therefore a more recent development than that used for CMIP5 experiments (Taylor et al., 2012), but an earlier version than the very recent LUH2 release (Hurtt et al., 2016). As LUH1 is a modeled product that is based on the underlying HYDE database, these products are very similar when the corresponding versions of each dataset are considered (Hurtt et al., 2011). Note that the version of HYDE used for our study (version 3.1.1, see above) is not the same as the version of HYDE that underlies the LUH1 data used here (early version of HYDE 3.2); the HYDE and LUH1 data used in this study differ in several aspects.*

revised Table 1:
Table 1. Overview of LPJ-GUESS simulations carried out as part of this study.

| Land-use model | | First year | Last year | Representation of LUC transitions | Spatial coverage |
|---|---|---|---|---|---|
| Abbreviation | Reference | | | | |
| LUH1 | 1500-2005: LUH1 (Hurtt et al., 2011), extension until 2014: Le Quéré et al. (2015b)[a] | 1700[b] | 2014 | gross | global |
| | | 1700[b] | 2014 | net | global |
| | | 1700[b] | 2014 | LUC fixed to 1700 | global |
| RAMA | 1700-1992: Ramankutty and Foley (1999), extension until 2007: Ramankutty (2012) | 1700 | 2007 | net | global |
| | | 1700 | 2007 | LUC fixed to 1700 | global |
| HYDE | 10 000 BC to AD 2000: HYDE3.1.1 (Klein Goldewijk et al., 2010, 2011), extension until 2005: see Klein Goldewijk et al.( 2015), extension until 2013: see Le Quéré et al. (2015a) | 1700[b] | 2013 | net | global |
| | | 1700[b] | 2013 | LUC fixed to 1700 | global |
| LUH2 | 850-2015: LUH2 v2 (release 14 Oct 16), (Hurtt et al., 2016) | 1700[b] | 2015 | net | global |
| | | 1700[b] | 2015 | LUC fixed to 1700 | global |
| HILDA | HILDA v2.0 Fuchs et al. (2015b) | 1900 | 2010 | gross | EU27[bc]+CH |
| | | 1900 | 2010 | net | EU27[c]+CH |
| | | 1900 | 2010 | LUC fixed to 1900 | EU27[c]+CH |
| LUH1 | 1500-2005: LUH1 (Hurtt et al., 2011), extension until 2014: (Le Quéré et al., 2015b) | 1900[d] | 2014 | net | EU27[c]+CH |
| | | 1900[d] | 2014 | LUC fixed to 1900 | EU27[c]+CH |

[a]Note that this version of LUH1 is based on an early version of HYDE 3.2, which is different from HYDE version 3.1.1 as used below, see methods. [b]1700 was selected as earliest start year, [c]EU 2007-2013, [d]1900 was selected as start year for European simulations.

*2. In the abstract the authors state that the main reason that gross land-use transitions have previously not been included in carbon modeling studies is the lack of detailed information on historical gross land-use changes. However, I would also argue that until recently many carbon models were not able to use even the simple gross land-use changes provided by land-use datasets.*

This is indeed an additional factor which was so far only picked up in the conclusions. We changed the abstract accordingly, which reads now:
abstract, line 23-26: *These complex changes between classes within a gridcell have often been neglected in previous studies, and only net changes of land between natural vegetation cover, cropland and pastures accounted for, mainly because of a lack of reliable high-resolution historical information on gross land transitions, in combination with technical limitations within the models themselves.*

*3. Another clarification: the LUH dataset includes shifting cultivation only in some locations within the tropics (based on the map of Butler 1980). There are currently several places in the paper where it is implied that shifting cultivation occurs throughout the tropics.*

It is true that shifting cultivation occurs not throughout the entire tropical area but is restricted to specific locations within the tropics according to implementation in the LUH dataset which is based on the map of Butler 1980. We changed the wording at several occasions in the text to "*certain tropical regions*" to be clear that the extent is not the entire tropical region (lines 31, 106, 123, 160, 169, 407, 412, 534,649) and inserted a link to the Butler map at the first occurrence of the expression.

*4. Although the authors use both net and gross land-use transitions in this study, they do not describe how they determine the net transitions for the LUH dataset (which by default provides gross transitions) or the HYDE dataset (which does not provide transitions at all – just land-use states). The calculation of net transitions should not be difficult in either case, but a brief description should be included for completeness in the methods section.*

The missing information was added to the methods section:

§2, lines 255-261: *LPJ-GUESS uses annual land use states of the classes cropland, pasture, natural vegetation and barren land (no vegetation, e.g. water or ice covered) as input for net LUC runs, that are complemented for gross LUC runs by annual gross transitions for each combination of two land-use classes. Land-use states of RAMA, HYDE and LUH2 were used directly. To generate net transitions from LUH1, annual land-use states were derived from land use states in 1700 and gross transitions from 1700 to 2014. HILDA land-use matrices providing land-use states and transitions together in form of an integer land-use category were translated to annual land-use states and gross transitions for each combination of two land-use classes.*

*5. The lack of wood harvest is a limitation of the modeling approach used by the authors and it would be good to include some more discussion of this. Wood harvest is one of the largest land-use transitions in terms of both area and carbon emissions. Although the spatial pattern of wood harvest is uncertain, national data on wood harvest amounts and areas are available. When comparing the effects of including only net vs. gross transitions it is important to consider that wood harvest is a gross transition that is currently not included in this study.*

We agree with the reviewer that wood harvest is an important form of managing natural resources, accounting for intense land-use transitions and carbon emissions. The uncertainty in the actual spatial pattern of wood harvest and the many existing ways wood harvest is done in practice over the globe (clear cut, selective harvesting of specific age classes or a mixture of both) introduces many possibilities as to how this process can be implemented in DGVMs. In an LPJ-GUESS-type of model where forest ecosystem and wood parameters vary significantly over tree age classes, this would result in a wide span of possible solutions depending on the parameters used for implementation of wood harvest. For this reason, we decided not to include one or more representation of wood harvest in our analysis as is stated in the introduction (lines 121-123), but in fact cite previous studies that assessed carbon emissions from wood harvest, sometimes in combination with the effects of shifting cultivation in the discussion (Houghton et al., 2012; Shevliakova et al., 2009, 2013; Wilkenskjeld et al., 2014). We feel that for a good estimate of carbon emissions from wood harvest an individual study would be necessary, allowing consideration of a variety of reasonable and technically possible implementations with a DGVM that represents age classes such as LPJ-GUESS.

To address the comment of the reviewer we added in the introduction that wood harvest is a form of forest management that can be represented as gross land transitions (lines 122-126) and added an explanation on the uncertainty that would come with an assessment of wood harvest with an LPJ-GUESS-type of model to the discussion section (lines 549-555).

introduction, lines 125-129: *We exclude wood harvest as a form of forest management that can be represented as gross land transitions from our analysis as, although national data on wood harvest are available, its parameterization in models is poorly constrained on a global scale, e.g. the effects strongly depend on assumptions on the harvest type (clear cut, selective logging, or a mixture of both), or assumptions regarding turnover times of harvested C (Wilkenskjeld et al., 2014).*

discussion, lines 600-607: *We did not account for wood harvest in this study as the uncertainty in the actual spatial pattern of wood harvest in combination with the ways wood harvest is done in practice over the globe (clear cut, selective harvesting of specific age classes or a mixture of both) introduces many possibilities as to how this process can be implemented in DGVMs. In a model such as LPJ-GUESS, where forest ecosystem and wood parameters vary significantly over tree age classes, this would result in a wide span of possible solutions depending on the parameters used for*

*implementation of wood harvest that would be better addressed in a thorough sensitivity study investigating a variety of possible implementations, rather than with a single representation.*

*6. The Discussion section begins by stating that a key uncertainty in estimating C stocks and fluxes from land-use stems from the choice of LUC dataset used. I think it would be good to rephrase this opening statement slightly to remind readers that historical reconstructions of land-use are inherently uncertain, and it is not just the choice of LUC dataset that introduces uncertainty. For example, \*all\* LUC datasets used in this study show a peak in LU transitions around 1950-1960, although there is some evidence that this is likely due to the reconstruction process itself (i.e. the merging of two or more data sources during that time period).*

Also here, the reviewer raises a valid point. It is true that LU datasets are inherently uncertain as a consequence of the limited data base on historical land use and land use transitions that need underlying model assumptions. We changed the statement accordingly. It reads now:

discussion, lines 433-436: *Resulting from the fact that historical reconstructions of land use and its changes are inherently uncertain because of the limited existing data base that needs complementary assumptions (e.g. on land rotation times), it is widely acknowledged that a key uncertainty in estimating changes in C stocks and fluxes as a response to LUC change stems from the choice of the LUC dataset (e.g. Houghton et al., 2012; Jain et al., 2013).*

*7. In the Conclusion section the authors state that the consideration of multiple LUC reconstructions exploring the full range of reasonable assumptions is needed. This was actually a central component of the paper of Hurtt et al. 2011 in which those authors performed a large sensitivity study by varying all model inputs and decision parameters to explore a range of possible land-use reconstructions.*

We added a link to this publication in the sentence.

conclusions, lines 630-635: *To account for the uncertainty arising from different reconstructions of historical LUC in the dynamic modeling of C stocks and fluxes and to provide realistic estimates of this uncertainty for the land-use C flux, the consideration of multiple LUC reconstructions exploring the full range of reasonable assumptions is needed, as well as efforts to narrow the uncertainty in constructions of historical land use. Multiple LUC reconstructions were calculated by Hurtt et al. (2011), but the consequences of uncertainty in land-use transitions are not routinely explored by the carbon cycle community.*

*8. Also in the Conclusion section the authors state that the differences in C stocks and fluxes predicted by the HYDE and LUH datasets is surprising given that they are based on the same data inputs etc. However, it appears that two different (inconsistent) versions of these datasets were used – see comment 1 above.*

See reply to comment 1 above for clarification on datasets used in the study. We excluded the statement from the discussion as indeed different versions were used for LUH and HYDE, so that differences are not surprising. We added a statement on the difference between the, now 4, datasets related to the date of their preparation.

discussion, line 621-625: *In our global simulations, HYDE/ RAMA and LUH1/LUH2 data often lead to similar results in ecosystem C stocks and fluxes. Therefore LUH1 and LUH2 as the more recent developments under the four considered reconstructions (both based on HYDE version 3.2, however*

*LUH1 on an intermediate version and LUH2 on the final version) differ more from older developments than these from each other (see methods for model versions used in this study).*

---

## Author Comment (AC2) · 30 Nov 2016

**Response to Interactive comment to anonymous referee #2**

Thank you for the positive assessment and suggestions to improve the manuscript. Below are the questions and suggestions followed by answers including the changes that were made in the manuscript.

In addition to addressing the reviewers comments we extended the analysis to including the newly available LUH2 net dataset in the analysis. Since this dataset will be used e.g. in CMIP6 simulations, we expect this dataset to emerge as one that will frequently be used for modeling studies in the future. As this we included it in the analysis. The inclusion of LUH2 in averages and uncertainties in C stocks and fluxes did not change the outcomes of this study. We also include a section on the inclusion of LUH2 as a model that is indeed not completely independent from LUH1 and HYDE in the discussion.

*1. Including in the discussion a critique of how wood harvest can be included in the future, and what sorts of feedbacks would be expected on the component flux emissions*

We included in the discussion section a paragraph on wood harvest and why we did not include it in our analysis.

discussion, §4.3, lines 600-607: *We did not account for wood harvest in this study as the uncertainty in the actual spatial pattern of wood harvest in combination with the ways wood harvest is done in practice over the globe (clear cut, selective harvesting of specific age classes or a mixture of both) introduces many possibilities as to how this process can be implemented in DGVMs. In a model such as LPJ-GUESS, where forest ecosystem and wood parameters vary significantly over tree age classes, this would result in a wide span of possible solutions depending on the parameters used for implementation of wood harvest that would be better addressed in a thorough sensitivity study investigating a variety of possible implementations, rather than with a single representation.*

In the same sense, emissions from various pools and the types of process feedbacks are dependent on the selected implementation and should be investigated in a separate study. As other studies showed, the consideration of wood harvest increases the annual land use flux (Houghton et al., 2012; Shevliakova et al., 2009). Emissions from transitions of forest to non-forest LU types would then be smaller (as average forest biomass would be lower when wood harvest is considered). Therefore especially the timing when fluxes from wood harvest are emitted to the atmosphere is affected by the ex-/inclusion of wood harvest. Likewise we would expect the distribution of soil legacy effects to differ. As our study does not address the issue of wood harvest, and because of the many uncertainties and technicalities involved in its implementation, we do not feel it appropriate for us to speculate here on the best way to include wood harvest. This should instead be the topic of a separate analysis that fully investigates the consequences of the various assumptions.

*2. Is forest age structure included in the model? The authors mention that 'young stands are harvested before old stands'. This implies there is age structure. Also, what is the logic for harvesting the young stands first rather than the old stands? What would be the implications of doing this in reverse?*

The LPJ-GUESS model is characterized through the representation of gap dynamics and age classes in cohorts of 5 years (Smith et al., 2001). This includes an explicit representation of vegetation structure, dynamics and competition between the individual cohorts of a PFT population. Replicate patches are simulated to take account of stochastic processes of individuals (establishment and mortality).

In the first version of the manuscript we have not mentioned the explicit representation of vegetation dynamics and age structure in the text. We corrected for this and added the following text:

§2.2, lines 212-2213: *Vegetation structure, dynamics and competition between age cohorts of a PFT population are explicitly represented in the LPJ-GUESS model.*

The assumption that, upon conversion of natural to agricultural lands, young stands are harvested before old stands, follows the idea that younger and not-yet-mature stands in practice would be cut down before old-growth forest would be touched. This assumption is not based on solid evidence, however, follows logical considerations that land managers might take and was applied in previous modeling studies e.g. (Hurtt et al., 2011; Stocker et al., 2014). It also reflects the likely accessibility of land where the forest has already been cut, versus land with old-growth forest. Doing the reverse, i.e. cutting down oldest stands first, would result in a probably unrealistically high immediate and long-term loss of above and below-ground C stocks, and therefore higher total emissions from land-use change.

*3. Fire is not discussed in the manuscript – how would the authors plan to include fire feedbacks and the reorganization of forest structure in their sub-grid cell based transitions?*

Fire is represented explicitly in the LPJ-GUESS model, with a full description of the algorithm given by Thonicke et al. (2001). Thus the forest structure simulated is an emergent result of a number of processes, including fire. For any full details on the model's structure and implementation (with fire among this) the reader is referred to the given references (Lindeskog et al., 2013; Sitch et al., 2003; Smith et al., 2014, 2001). As such, and because fire is not a focus of the manuscript, fire is not mentioned explicitly.
The status of global fire modeling is discussed in a recent publication by Hantson et al. (2016). It shows that there are advances in representing fires in global models but also a number of challenges attached to the process of finding reliable representations and predictions. Fire dynamics will be influenced under land rotations such as shifting cultivation, and are certainly one aspect that could be the focus of a dedicated study. Since this is not within the scope of the manuscript, we do not treat this issue in any detail but added a statement to the discussion that relates to different vegetation and ecosystem dynamics under repeated land changes.

discussion §4.3, lines 596-599: *In the same way, other land-use and land-cover change-related processes (e.g. fate of harvested wood, residue management, occurrence of disturbances, including fire, etc.) might differ under repeated land transitions such as shifting cultivation, and realistic representation of these interactions in process-based models remains a subject for further research.*

**References**

Hantson, S., Arneth, A., Harrison, S.P., Kelley, D.I., Prentice, I.C., Rabin, S.S., Archibald, S., Mouillot, F., Arnold, S.R., Artaxo, P., Bachelet, D., Ciais, P., Forrest, M., Friedlingstein, P., Hickler, T., Kaplan, J.O., Kloster, S., Knorr, W., Lasslop, G., Li, F., Mangeon, S., Melton, J.R., Meyn, A., Sitch, S., Spessa, A., van der Werf, G.R., Voulgarakis, A., Yue, C., 2016. The status and challenge of global fire modelling. Biogeosciences Discuss. 1–30. doi:10.5194/bg-2016-17

Houghton, R.A., House, J.I., Pongratz, J., van der Werf, G.R., DeFries, R.S., Hansen, M.C., Le Quéré, C., Ramankutty, N., 2012. Carbon emissions from land use and land-cover change. Biogeoscience 9, 5125–5142. doi:10.5194/bg-9-5125-2012

Hurtt, G.C., Chini, L.P., Frolking, S., Betts, R.A., Feddema, J., Fischer, G., Fisk, J.P., Hibbard, K., Houghton, R.A., Janetos, A., Jones, C.D., Kindermann, G., Kinoshita, T., Klein Goldewijk, K., Riahi, K., Shevliakova, E., Smith, S., Stehfest, E., Thomson, A., Thornton, P., van Vuuren, D.P., Wang, Y.P., 2011. Harmonization of land-use scenarios for the period 1500–2100: 600 years of global gridded annual land-use transitions, wood harvest, and resulting secondary lands. Clim. Change 109, 117–161.

Lindeskog, M., Arneth, A., Bondeau, A., Waha, K., Seaquist, J., Olin, S., Smith, B., 2013. Implications of accounting for land use in simulations of ecosystem carbon cycling in Africa. Earth Syst. Dyn. 4, 385–407. doi:10.5194/esd-4-385-2013

Shevliakova, E., Pacala, S.W., Malyshev, S., Hurtt, G.C., Milly, P.C.D., Caspersen, J.P., Sentman, L.T., Fisk, J.P., Wirth, C., Crevoisier, C., 2009. Carbon cycling under 300 years of land use change: Importance of the secondary vegetation sink. Global Biogeochem. Cycles 23, 1–16. doi:10.1029/2007GB003176

Sitch, S., Smith, B., Prentice, I.C., Arneth, A., Bondeau, A., Cramer, W., Kaplan, J.O., Levis, S., Lucht, W., Sykes, M.T., Thonicke, K., Venevsky, S., 2003. Evaluation of ecosystem dynamics, plant geography and terrestrial carbon cycling in the LPJ dynamic global vegetation model. Glob. Chang. Biol. 9, 161–185.

Smith, B., Prentice, I.C., Sykes, M.T., 2001. Representation of vegetation dynamics in the modelling of terrestrial ecosystems: comparing two contrasting approaches within European climate space. Glob. Ecol. & Biogeogr. 10, 621– 637 10, 621–637.

Smith, B., Wårlind, D., Arneth, A., Hickler, T., Leadley, P., Siltberg, J., Zaehle, S., 2014. Implications of incorporating N cycling and N limitations on primary production in an individual-based dynamic vegetation model. Biogeosciences Discuss. 11, 2017–2054. doi:10.5194/bgd-10-18613-2013

Stocker, B.D., Feissli, F., Strssmann, K.M., Spahni, R., Joos, F., Strassmann, K., 2014. Past and future carbon fluxes from land use change, shifting cultivation and wood harvest. Tellus B 1, 1–15. doi:10.3402/tellusb.v66.23188

Thonicke, K., Venevski, S., Sitch, S., Cramer, W., 2001. The role of fire disturbance for global vegetation dynamics: coupling fire into a Dynamic Global Vegetation Model. Glob. Ecol. Biogeogr. Lett. 10, 661–678.

Wilkenskjeld, S., Kloster, S., Pongratz, J., Raddatz, T., Reick, C., 2014. Comparing the influence of net and gross anthropogenic land use and land cover changes on the carbon cycle in the MPI-ESM. Biogeosciences Discuss. 11, 5443–5469. doi:10.5194/bgd-11-5443-2014

---

## Author Comment (AC3) · 30 Nov 2016

Thank you for the comments and suggestions for our manuscript. The manuscript became much clearer to follow in many of the sections based on your suggestions.

Please also note the supplement to this comment:
http://www.earth-syst-dynam-discuss.net/esd-2016-24/esd-2016-24-AC3-supplement.pdf

---

## Author Response (AR1)

Dear Christian Reick,

Thank you for the offer to resubmit the revised manuscript. Please note that one further minor changed has been applied.

on line 221, "10% of leaf biomass", has been changed to, "71% of leaf biomass". This error was noted after submission of the response to reviewers, but is purely descriptive in nature, and does not affect the results in any way.

Best,

Thomas Pugh, Anita Bayer, et al.

[revised manuscript text omitted]

---

## Editor Decision (ED1)

**Editor's comment on**

**A. Bayer, M. Lindeskog, T.A.M. Pugh, P. Anthoni, R. Fuchs and A. Arneth**

**"Uncertainties in the land use flux resulting from land use change reconstructions and gross land transitions"**

**Earth System Dynamics Discussions, doi:10.5194/esd-2016-24**

December 22, 2016

As editor I thank the authors for carefully addressing the issues raised by the reviewers. In particular I appreciate the inclusion of the LUH2 dataset into the revised manuscript.

For preparing the final version of your paper, I would like to point your attention to the paper by Hansis et al. (2015) that also addresses several types of uncertainties (harvest, legacy fluxes, initial state). I suggest to compare your numbers for uncertainties with theirs in the discussion.

MINOR REMARKS:

- Line 137: Since the datasets are so basic for your paper, I suggest you shift table S1 listing the dataset characteristics from the supplement to the paper itself. Or you merge it with table 1 that contains overlapping information.

- Lines 178/179: Would it be useful to name explicitly in which "several aspects" the data sets differ?

- Line 249: What is the meaning of the abbreviation "CFT"?

- Line 644: Uncertainties in emissions from land use change arising from different grid resolutions have been estimated in the paper by Wilkenskjeld et al. (2014) that you already cite in different contexts.

- Figs. 1 and 2: The grey color in your plots is hardly recognizable. I suggest to use another color.

- All Figures: Please consider whether you want to use a larger font in the figures, it may happen that in the final paper the figures are so small that labels are hardly readable.

LITERATURE:

Hansis, E., S. J. Davis, and J. Pongratz (2015), Relevance of methodological choices for accounting of land use change carbon fluxes, Global Biogeochem. Cycles, 29, 12301246, doi:10.1002/2014GB004997.

I wish a happy Christmas!

Christian Reick

---

## Author Response (AR2)

**Response to editor**

We would like to thank Dr Reick for the useful suggestions to improve our manuscript. The changes we have made in response to those suggestions are outlined below. A few grammatical corrections have also been made on further reading. These are highlighted in red in the revised manuscript.

5 **For preparing the final version of your paper, I would like to point your attention to the paper by Hansis et al. (2015) that also addresses several types of uncertainties (harvest, legacy fluxes, initial state). I suggest to compare your numbers for uncertainties with theirs in the discussion.**

Thanks for alerting us to this. The following two references to Hansis et al. have been added.

Line 526-527:
10 "Using a bookkeeping model, Hansis et al. (2015) found an enhancement of $E_{LUC}$ by 22-24% over the period 1500-2012, at a 0.5° x 0.5° spatial resolution."

Line 625-632:
"Hansis et al. (2015) in contrast found a 28% higher $E_{LUC}$ over the period 2008-2012 when legacy emissions predating 2008 were excluded. This resulted from previous LUC transitions generating
15 more uptake from regrowth than loss from soil decomposition during the 2008-2012 period, and further exemplifies the considerable importance of legacy effects for calculation of instantaneous emissions. They also emphasize that the change in soil and vegetation stocks induced by previous LUC can substantially modify $E_{LUC}$ calculations compared to an assumption of equilibrium in the initial stocks. Often such changes in initial conditions tend to lower vegetation C stocks and thus
20 subsequent deforestation emissions. Our 17% lower $E_{LUC}$ excluding legacy emissions accounts for both these effects."

**Line 137: Since the datasets are so basic for your paper, I suggest you shift table S1 listing the dataset characteristics from the supplement to the paper itself. Or you merge it with table 1 that contains overlapping information.**

25 The previous Table S1 is now Table 1 of the revised manuscript. Overlapping information in the old Table 1 (now Table 2) has been removed.

**Lines 178/179: Would it be useful to name explicitly in which "several aspects" the data sets differ?**

The text has now been revised as follows:
" Note that the version of HYDE used for our study (version 3.1.1, see above) is not the same as the
30 version of HYDE that underlies the LUH1 data used here (early version of HYDE 3.2). HYDE 3.1.1 and 3.2 differ in terms of driving population data and the algorithms used (Klein Goldewijk et al., 2016). The HYDE and LUH1 data used in this study therefore differ in both their spatial and temporal distribution of land-use fractions (Fig. 1; Klein Goldewijk et al., 2016)."

**Line 249: What is the meaning of the abbreviation "CFT"?**

35 Crop Functional Type. This has now been defined in the text where it is first used in Section 2.2.

**Line 644: Uncertainties in emissions from land use change arising from different grid resolutions have been estimated in the paper by Wilkenskjeld et al. (2014) that you already cite in different contexts.**

We have added ", as shown by Wilkensjeld et al. (2014)" to the end of this sentence to reflect this.

40 **Figs. 1 and 2: The grey color in your plots is hardly recognizable. I suggest to use another color.**

**All Figures: Please consider whether you want to use a larger font in the figures, it may happen that in the final paper the figures are so small that labels are hardly readable.**

Figures have been revised as suggested.

45 Best regards,

Thomas Pugh (on behalf of Anita Bayer et al.)

[revised manuscript text omitted]